# The psychological distress of parents is associated with reduced linear growth of children: Evidence from a nationwide population survey

Kun A. Susiloretni[1]*, Emily R. Smith[2], Suparmi[3], Marsum[1], Rina Agustina[4,5], Anuraj H. Shankar[6,7]

1 Semarang Health Polytechnic Ministry of Health - Poltekkes Kemenkes Semarang, Semarang, Central Java, Indonesia, 2 Department of Global Health, George Washington University, Washington, D.C., United States of America, 3 National Health Institute Research and Development, Jakarta, Indonesia, 4 Faculty of Medicine, Department of Nutrition, Universitas Indonesia – Dr. Cipto Mangunkusumo General Hospital, Jakarta, Indonesia, 5 Faculty of Medicine, Human Nutrition Research Center, Indonesian Medical Education and Research Institute, Universitas Indonesia, Jakarta, Indonesia, 6 Eijkman-Oxford Clinical Research Unit, Eijkman Institute for Molecular Biology, Jakarta, Indonesia, 7 Nuffield Department of Medicine, The Centre for Tropical Medicine and Global Health, University of Oxford, Oxford, United Kingdom

* kun@poltekkes-smg.ac.id

**Data Availability Statement:** Data cannot be shared publicly because the data belong to the National Health Institute Research and

## Abstract

### Background

Stunting, an indicator of restricted linear growth, has become a primary measure of childhood undernutrition due to its persistent high prevalence globally, and importance for health and development. Although the etiology is recognized as complex, most analyses have focused on social and biomedical determinants, with limited attention on psychological factors affecting care and nurturing in the home. We assessed whether the psychological distress of parents is related to child linear growth and stunting, and documented the associated risk factors, and examined the relationship between parental distress and behavioral and other risk factors for stunting.

### Methods

We used data from the Indonesia National Health Survey 2013, including 46,315 children 6–59 months of age. Multivariate linear, logistic, and multilevel multinomial logistic regression, using survey weights, were used to assess the relationship between parental distress, as assessed by the WHO Self Reporting Questionnaire (SRQ20), with height-for-age z score (HAZ), stunting, and behavioral and other risk factors for stunting.

### Results

Maternal, paternal and parental distress (i.e. both maternal and paternal distress) were associated with reduced linear growth of the children by 0.086 (95% CI -0.17, -0.00), 0.11 (95% CI -0.24, -0.02) and 0.19 (95% CI -0.37, -0.00) HAZ-scores, respectively. Maternal and paternal distress increased the risk of mild stunting (HAZ <-1) by 33% (95% CI

Development (NHIRD) Ministry of Health Republic Indonesia and authors do not have permission to share the data. Data are available from the NHIRD Institutional Data Access (contact via http://labmandat.litbang.kemkes.go.id/menu-layan) for researchers who meet the criteria for access to confidential data. The authors did not receive any special privileges in accessing the data from the NHIRD.

**Funding:** No funding sources. None of the authors were given an honorarium, grant, or other form of payment.

**Competing interests:** The authors have declared that no competing interests exist.

1.17,1.50) and 37% (95% CI 1.18,1.60), and the risk of moderate stunting (HAZ <-2) by 25% (95% CI 1.10,1.43) and 28% (95% CI 1.08,1.51]), respectively. Parental stress increased the risk of moderate stunting by 40% (95% CI 1.06,1.85). Amongst specific groups of risk factors, the proportion of HAZ-score lost was associated with socioeconomic factors (30.3%) including, low wealth, low maternal occupational status, low maternal education, rural residence, and low paternal occupational status; physiological factors (15.5%) including low maternal height, low maternal mid-upper arm circumference, being male, low paternal height; behavioral factors (8.9%) including open garbage disposal, paternal smoking, not using iodized salt; and experiencing at least one infectious diseases episode (1.1%).

## Conclusions

Maternal, paternal and parental stress were associated with reduced linear growth of children. These findings highlight the complex etiology of stunting and suggest nutritional and other biomedical interventions are insufficient, and that promotion of mental and behavioral health programs for parents must be pursued as part of a comprehensive strategy to enhance child growth and development, i.e. improved caretaker capacity, integrated community development, improved parenting skills, as well as reduced gender discrimination, and domestic violence.

## Introduction

Linear growth restriction during childhood has profound impacts on health throughout the human life course. Short-term effects include increased mortality, morbidity, and disability. Long-term consequences include reductions in adult size, low economic productivity, reduced reproductive performance, impaired intellectual ability, and increased risk of metabolic and cardiovascular disease [1–4]. Stunting, an indicator of restricted linear growth, has become a primary focus of childhood undernutrition due to its persistent high prevalence globally, and importance for health and development [4,5]. Multiple studies of stunting have been conducted, and repeatedly identified the risk factors of low maternal stature, low birth weight, childhood gastrointestinal infections, and low socioeconomic status [6]. In accord with birth weight, a recent study added fetal growth restriction and preterm birth as risk factors and suggested they account for as much as 32% of stunted children [6]. Additional risks in some studies include environmental factors [7–9], maternal health status [10–12], and child health status [13,14]. However, these risk factors combined are still insufficient to explain the current levels of stunting. Thus, the etiology of restricted linear growth warrants deeper exploration [4,6,15].

Few studies have explored the impact of household socio-emotional risk factors on growth restriction. Factors such as parental psychological distress could diminish the quality of caregiving behaviors and enhance psychological stress for the child, both of which may affect growth via the hypothalamic–pituitary–adrenal (HPA) axis and other pathways [16,17]. Moreover, the stress in pregnant women and children could influence nutrient metabolism or immune function, further limiting fetal and child growth and adversely affecting health [18,19].

The current Covid-19 pandemic has affected psychological distress for parents which is associated with parenting difficulties including harsh parenting [20], child maltreatment [21], and food insecurity [22]. A nationwide study in China with 52,730 participants shows that

35% had experienced pandemic-related psychological distress [23]. A survey in the US reported serious psychological distress was only 3.9% in 2018 compared to 13.6% in 2020 [24]. This increase would likely exacerbate existing vulnerabilities for child growth and development.

Previous studies of psychological distress assessment in the low and middle-income countries (LMIC) of India and Vietnam using the Self Reporting Questionnaire (SRQ20), found that high maternal mental disorder contributed to child stunting and underweight status. Another study in Bangladesh and Vietnam where poverty, malnutrition, and poor mental health coexist, found maternal mental disorder was related to child stunting and underweight [25,26]. In addition, previous research in Brazil found that BMI-for-age z-scores of children were negatively associated with maternal mental disorder scores at 5–8 years postpartum [27]. However, these studies were of either limited size or conducted predominantly in groups experiencing elevated stress levels, and they did not collect data on many of the other known predictors of poor linear growth. As such, their findings may have limited relevance to the general population and are subject to confounding.

To address this gap in knowledge of causes of growth restriction, and to better understand links between parental stress and child linear growth, we examined data from Indonesia where the prevalence of stunting over the last 10 years has remained between 33.6% to 37.2%, and the prevalence of parental mental disorders between 6.0% and 11.6% as indicated by the Indonesia National Health Survey (INHS) from 2007, 2010, 2013 and 2016 [28,29]. Specifically, we utilized the INHS 2013 national-level dataset with complete data on psychological distress in parents, child height, and other nutritional, behavioral, and social factors associated with stunting. We examined the association between parental distress and child linear growth and stunting, and calculated height-for-age z score loss due to parental distress and socioeconomic, behavioral, and physiologic risk factors for linear growth.

## Materials and methods

### Design

We used nationally representative data from the INHS 2013. This was cross-sectional household survey data, designed to be representative at the national, provincial, and district level. The sampling frame consisted of 12,000 census blocks selected using probability proportional to population from 30,000 Primary Sampling Units (PSUs). There were 294,959 households visited (98.3%) of 300,000 households targeted, from 33 provinces and 497 districts/cities. The INHS 2013 collected more than 1,000 variables using household and individual questionnaires. Data collection was in 2013 and approved by the Ethics Committee for Health Research of the National Institute of Health Research and Development (NIHRD) of the Ministry of Health of the Republic of Indonesia. All participants gave written consent [28]. We selected data from all households comprised of two-parent families with children age 6 to 59 months, and with data on parental distress. The dataset included all 6 to 59 months old children living in the household.

### Variables

The primary outcomes of this study were linear growth and stunting prevalence. Height and length were measured to the nearest 0.1 cm [28], and age was assessed from respondent statements of date of birth confirmed with legal documents such as household ID listing and birth certificate. These were used to calculate the height-for-age z-score (HAZ) using WHO Anthro version 3.2. software [30]. We excluded children who were missing anthropometric measurements as this precluded calculation of HAZ, weight-for-age z-score (WAZ), and weight-for-height z-score (WHZ). We also excluded children with extreme values for HAZ resulting from

errors in measurement or data entry, if HAZ was below –6 or above +6; and to further optimize data quality we excluded children if WAZ was below –6 or above +5, or WHZ was below –5 or above +5, BMIZ was below –5 or above +5 as these utilized either height or age, possibly affecting accuracy of the HAZ score [31,32]. Because the hazardous effects of risks for stunting could occur along a continuum of mild, moderate, and severe malnutrition [33,34], we included mild stunting ($< -1$ HAZ), moderate ($< -2$ HAZ) and severe ($< -3$ HAZ) stunting as outcomes to assess associations of risk factors with restricted linear growth.

Parental psychological distress was assessed using the Self Reporting Questionnaire (SRQ20) developed for adults by WHO. The SRQ20 is comprised of 20 questions related to neurotic symptoms [35] and has been validated in many developing countries, including Malaysia, The Philippines, and India [36–38]. Mothers and fathers were asked to indicate whether they had experienced any of the 20 symptoms ('yes' = 1 or 'no' = 0). We calculated maternal and paternal distress scores by summing the item scores, which ranged from 0 to 20 (0 indicating 'no distress' to 20 indicating 'severe distress'), and categorized maternal and paternal distress into a binary variable using a cutoff point of 6 [28]. A cutoff score <6 indicated 'no distress', and $\geq$6 indicated 'distress'. We created four parental distress categories: both the mother and father with no distress (no distress), only the mother with distress (maternal distress), only the father with distress (paternal distress), and both the mother and father with distress (parental distress).

Other covariates of interest included infectious disease, physiological, health behavior, and socioeconomic factors of the child and parents. Infectious disease factors consisted of binary variables indicating whether or not the child suffered from diarrhea, acute respiratory infection, pneumonia, or malaria in the last month. Physiological factors were mid-upper arm circumference (MUAC), body mass index (BMI), height and age of the mother, height and age of the father, and child sex. Health behavior factors consisted of the use of iodized salt, drinking water source, garbage disposal, water waste disposal, stool disposal, handwashing behavior, attending growth monitoring services, and paternal smoking. Social factors were education, occupation of mother and father, number of household members, rural or urban residence, and wealth quintile (Fig 1).

## Statistical analyses

Descriptive statistics were tabulated to assess the distribution of respondents' characteristics by each variable. The multivariate-adjusted analyses incorporating sampling weights were conducted using linear, logistic, and multilevel multinomial logistic regression with PSU as a random effect. We explored inclusion of a random effect for household nested within PSU as 20% of households had more than one child 6 to 59 months old; however, this did not change beta coefficients nor p values and was therefore not maintained in the analysis. We determined the full and the best fitting model for the relative risk of stunting associated with parental distress and other covariates. We present the beta coefficients for association with HAZ-score and proportion of HAZ-score lost due to parental distress and other risk factors. For the proportion of z-score lost, the numerator was the sum of the products for each child of the beta coefficient of interest that was significant times the variable value, and the denominator was the sum of the products for each child of all beta coefficients of that were significant times the variable value plus the value of the intercept.

$$\% \text{ HAZ score lost for risk factor Y} = \frac{\text{sum of the product for each child for significant risk factor Y beta} * \text{Y variable value}}{\text{sum of the product for each child for all significant risk factor betas} * \text{variable values} + \text{intercept}} * 100$$

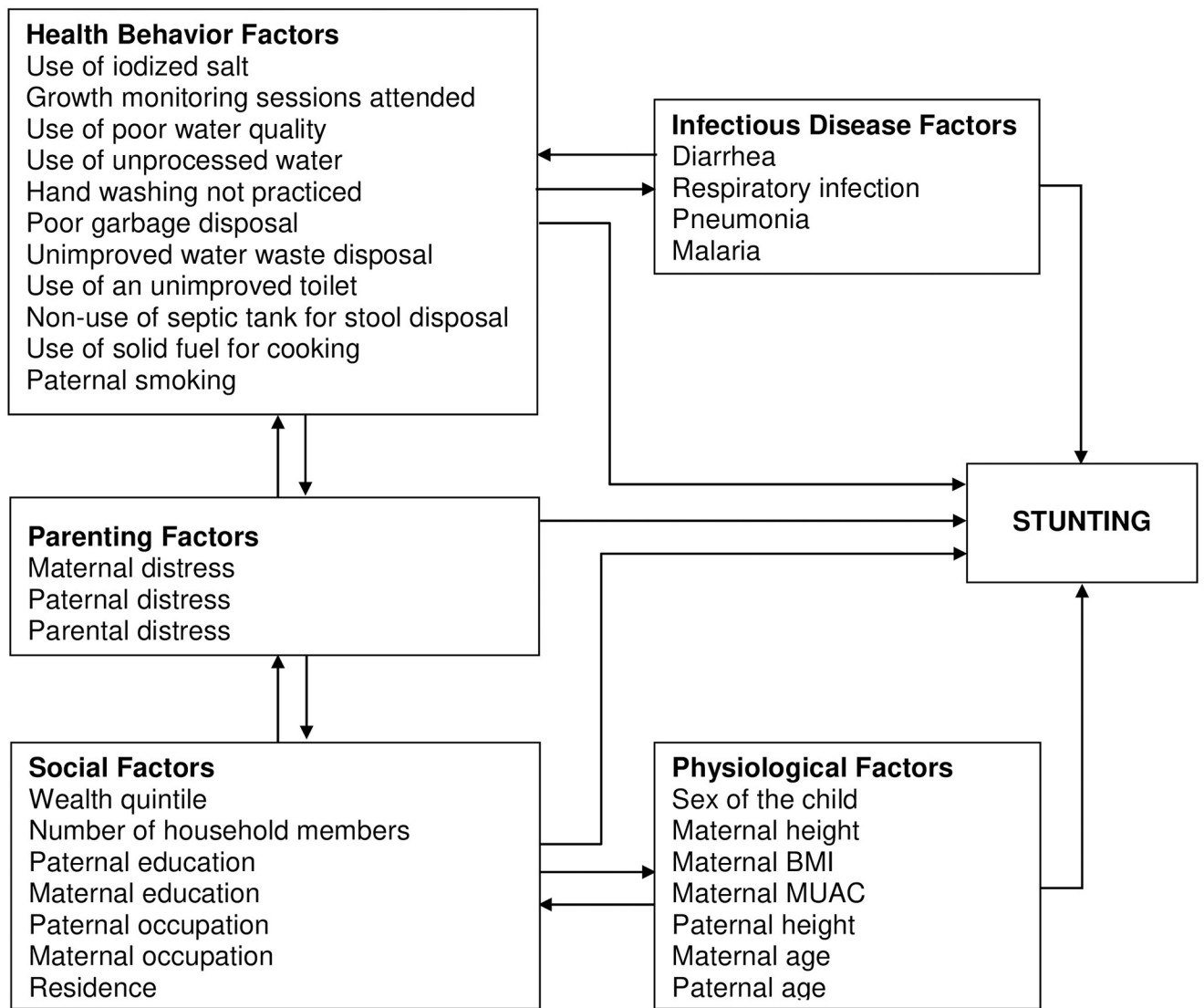

**Fig 1. Conceptual framework of the impact of distress of parents and other factors on child stunting.** The various groups of factors interact directly and indirectly with each other and with distress, and indirectly and directly on stunting.

We also examined the relationship between parental distress and behavioral risk factors for stunting using multilevel logistic and multinomial logistic regression. All analyses were run using STATA 15 of StataCorp. College Station, Texas.

## Results

The flow diagram of the study participants is shown in Fig 2. From a total of 75,440 households with children 6–59 months of age, there were 46,315 households with complete parent and anthropometric data eligible for analysis. The mean (SD) age of the children was 35.2(±15.3) months; fathers and mothers were 36.1 (±7.3) years and 31.6 (±6.2) years old, respectively. More than half of mothers (59.9%) were not working, while almost all fathers (97.0%) were working. We observed that 28,726 (62,0%) and 26,975 (58.2%) of the mothers and fathers had

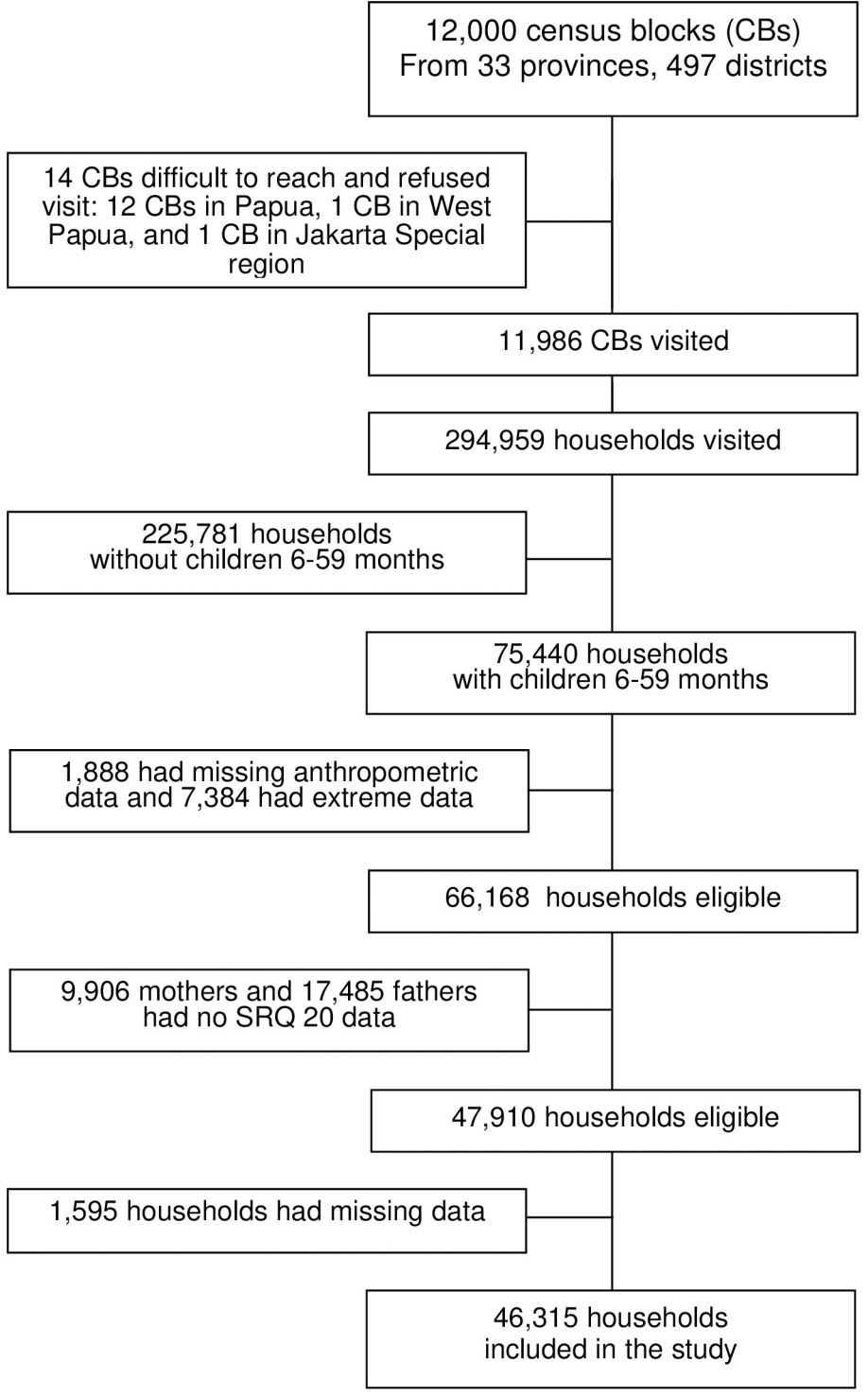

**Fig 2. Flow diagram of participant assessed.**

never attended a high school, respectively. The study households had a mean family size of five. Of the 46,315 children age 6–59 months, 63.8% overall were either mildly (26,6%), moderately (20,5%), or severely (16.7%) stunted. The mean HAZ was -1.39± 1,88. The proportion of mothers, fathers, and parents with distress (i.e. SRQ score of at least 6) were 4.1%, 2.6%, and 0.9%, respectively. Table 1 shows the distribution of variables included in the study and stratified by child stunting status (not stunted, mildly, moderately, and severely stunted).

As seen in Fig 3, higher scores of maternal, paternal, and parental distress were associated with lower HAZ scores (p<0.05). Table 2 presents the beta coefficients for HAZ of maternal, paternal, and parental distress and other covariates grouped by social, physiological, health behavior, disease, and parental distress factors. Maternal, paternal and parental distress were significantly related to lower HAZ scores by -0.09 (95%CI -0.17 to -0.00, p = 0.048), -0.11 (95% CI -0.21 to -0.00, p = 0.017), and -0.19 (95%CI -0.37 to -0.00, p = 0.045), respectively.

The relationship between distress of parents and stunting is depicted in Fig 4 along with other risk factors for stunting. In the best fitting model, distress was a significant risk factor for mild and moderate stunting. The relative risk ratios of maternal, paternal, and parental distress for mild stunting, respectively, were 1.33 (95%CI 1.17 to 1.50, p<0.001), 1.37 (95%CI 1.18 to 1.60, p<0.001), and 1.19 (95%CI 0.90 to 1.57, p = 0.228). For moderate stunting, the relative risk ratios, respectively, were 1.25 (95%CI 1.23 to 1.62, p = 0.001), 1.28 (95%CI 1.08 to1.51, p = 0.004), and 1.40 (95%CI 1.06 to 1.85, p = 0.019). For severe stunting, the relative risk ratios for maternal, paternal, and parental stress, respectively, were not significant at 0.93 (95%CI 0.80 to 1.09, p = 0.374), 1.00 (95%CI 0.83 to 1.20, p = 0.965), and 1.18 (95%CI 0.87 to 1.59, p = 0.298). The detailed tables of analyses are presented in the S1 Appendix.

Other factors significantly related to stunting included infectious disease, physiological factors, health behaviors, and socioeconomic factors. Infectious disease included morbidity from diarrhea, acute respiratory infections, and malaria. Children who experienced at least one infectious disease were at higher risk to be severely stunted. We note that infectious diseases were not associated with mild and moderate stunting.

Physiological factors related to stunting were child sex (boy), low maternal MUAC, older maternal and paternal age, and paternal or, maternal short stature. Boys tended to have a 7% higher risk of severe stunting compared to girls (RR 1.07; 95%CI 1.01 to 1.14; p = 0.033). Maternal and paternal physiological factors related to stunting included low maternal or paternal height, as well as low maternal MUAC. Maternal height less than 150cm was associated with a higher risk of mild (RR 1.49, 95%CI 1.41 to 1.57; p<0.001), moderate (RR 2.14, 95%CI 2.02 to 2.26; p<0.001) and severe (RR1.82, 95%CI 1.71 to 1.93; p<0.001) stunting. Paternal height less than 155cm was associated with mild (RR = 1.22, 95%CI 1.10 to 1.35; p<0.001), moderate (RR = 1.62, 95%CI 1.47 to 1.80; p<0.001) and severe (RR = 1.81, 95%CI 1.63 to 2.00; p<0.001) stunting. Maternal MUAC was also related to risk of mild (RR = 1.13, 95%CI 1.06 to 1.21; p<0.001), moderate (RR = 1.23, 95%CI 1.14 to 1.32; p<0.001) and severe (RR = 1.30, 95%CI 1.21 to 1.41; p<0.001) stunting.

Several behavioral factors related to stunting included non-use of iodized salt, poor garbage disposal, unimproved water waste disposal, and paternal smoking. However, households of children who used poor water quality, used unprocessed water, poor hand-washing behavior, non-use of toilets and septic tank for stool, and used solid fuel for cooking were not associated with stunting. Children in households with non-use of iodized salt had a higher risk for mild (RR 1.09; CI 95%1.02 to 1.16; p = 0.008) and moderate (RR 1.14, CI 95%1.06 to 1.22, p<0.001) stunting. Households who used poor garbage disposal had a higher risk for moderate (RR = 1.15, 95%CI 1.08 to 1.24; p<0.001) and severe (RR = 1.19, 95%CI 1.10 to 1.28; p<0.001) stunting. Children in households that used unimproved water waste disposal had a higher risk of mild (RR = 1.09, 95%CI 1.01 to 1.16; p = 0.022) and moderate (RR = 1.10, 95%CI 1.01 to

**Table 1. Characteristics of children and household according to stunting categories.**

| Risk factors | Normal | Mild stunting | Moderate stunting | Severe stunting | Total | |
|---|---|---|---|---|---|---|
| | % | % | % | % | n | % |
| Stunting | 36.2 | 26.6 | 20.5 | 16.7 | 46,315 | 100.0 |
| Child Age (months) | | | | | | |
| 6–11 | 4.1 | 1.7 | 1.1 | 1.3 | 3,821 | 8.3 |
| 12–23 | 7.1 | 4.4 | 3.7 | 3.7 | 8,708 | 18.8 |
| 24–35 | 7.2 | 5.2 | 4.4 | 3.9 | 9,575 | 20.7 |
| 36–47 | 8.6 | 6.9 | 5.3 | 4.0 | 11,553 | 24.9 |
| 48–59 | 9.1 | 8.5 | 6.0 | 3.7 | 12,658 | 27.3 |
| Distress of parent | | | | | | |
| No distress | 33.9 | 24.3 | 18.6 | 15.5 | 42,820 | 92.5 |
| Maternal distress | 1.2 | 1.3 | 1.0 | 0.6 | 1,885 | 4.1 |
| Paternal distress | 0.8 | 0.8 | 0.6 | 0.4 | 1,201 | 2.6 |
| Parental distress | 0.2 | 0.2 | 0.2 | 0.2 | 409 | 0.9 |
| Had infectious diseases | | | | | | |
| No disease | 25.3 | 18.4 | 13.9 | 11.4 | 31,941 | 69.0 |
| ≥1 disease | 10.9 | 8.2 | 6.6 | 5.4 | 14,374 | 31.0 |
| Sex of the child | | | | | | |
| Girl | 18.0 | 13.3 | 10.2 | 7.8 | 22,829 | 49.3 |
| Boy | 18.2 | 13.3 | 10.3 | 8.9 | 23,486 | 50.7 |
| Maternal BMI | | | | | | |
| Normal weight | 30.4 | 22.1 | 17.0 | 14.1 | 38,730 | 83.6 |
| Underweight | 2.3 | 1.9 | 1.6 | 1.4 | 3,313 | 7.2 |
| Obese | 3.5 | 2.7 | 1.8 | 1.2 | 4,252 | 9.2 |
| Maternal MUAC (cm) | | | | | | |
| ≥ 23.5 | 31.1 | 22.2 | 16.5 | 13.3 | 38,526 | 83.2 |
| < 23.5 | 5.1 | 4.4 | 3.9 | 3.4 | 7,789 | 16.8 |
| Maternal height (cm) | | | | | | |
| ≥ 150 | 27.5 | 17.8 | 11.7 | 10.2 | 31,158 | 67.3 |
| < 150 | 8.7 | 8.8 | 8.7 | 6.5 | 15,157 | 32.7 |
| Paternal height (cm) | | | | | | |
| ≥ 155 | 34.3 | 24.8 | 18.5 | 14.9 | 42,816 | 92.4 |
| < 155 | 1.9 | 1.8 | 2.0 | 1.8 | 3,499 | 7.6 |
| Maternal age (years) | | | | | | |
| <25 | 4.5 | 3.5 | 2.8 | 2.5 | 6,156 | 13.3 |
| 25–34 | 20.0 | 14.5 | 11.0 | 8.9 | 25,185 | 54.4 |
| ≥ 35 | 11.6 | 8.7 | 6.7 | 5.3 | 14,974 | 32.3 |
| Paternal age (years) | | | | | | |
| <30 | 6.4 | 4.6 | 3.6 | 3.3 | 8,265 | 17.8 |
| 30–39 | 18.9 | 13.7 | 10.6 | 8.4 | 23,970 | 51.8 |
| ≥ 40 | 10.8 | 8.3 | 6.3 | 5.0 | 14,080 | 30.4 |
| Iodized salt used | | | | | | |
| Yes | 29.7 | 21.2 | 15.9 | 13.1 | 37,053 | 80.0 |
| No | 6.5 | 5.4 | 4.5 | 3.6 | 9,262 | 20.0 |
| Growth monitoring attended in last six months (times) | | | | | | |
| = 6 | 25.3 | 19.0 | 14.8 | 11.8 | 32,799 | 70.8 |
| < 6 | 10.8 | 7.7 | 5.7 | 5.0 | 13,516 | 29.2 |
| Used poor water quality | | | | | | |

*(Continued)*

**Table 1.** (Continued)

| Risk factors | Normal | Mild stunting | Moderate stunting | Severe stunting | Total | |
|---|---|---|---|---|---|---|
| | % | % | % | % | n | % |
| No | 34.0 | 24.8 | 18.8 | 15.3 | 43,074 | 93.0 |
| Yes | 2.2 | 1.8 | 1.6 | 1.4 | 3,241 | 7.0 |
| Used unprocessed water | | | | | | |
| Yes | 24.4 | 18.9 | 15.1 | 12.3 | 32,757 | 70.7 |
| No | 11.8 | 7.7 | 5.4 | 4.4 | 13,558 | 29.3 |
| Washed hand | | | | | | |
| Yes | 23.8 | 16.9 | 12.6 | 10.1 | 29,353 | 63.4 |
| No | 12.4 | 9.7 | 7.9 | 6.6 | 16,955 | 36.6 |
| Poor garbage disposal | | | | | | |
| No | 10.5 | 8.1 | 6.3 | 5.4 | 14,022 | 30.3 |
| Yes | 25.7 | 18.5 | 14.2 | 11.3 | 32,293 | 69.7 |
| Used water waste disposal | | | | | | |
| Improved | 11.5 | 7.1 | 4.5 | 3.5 | 12,300 | 26.6 |
| Unimproved | 24.7 | 19.5 | 16.0 | 13.2 | 34,015 | 73.4 |
| Used a toilet | | | | | | |
| Improved | 27.7 | 19.3 | 13.7 | 10.7 | 33,044 | 71.3 |
| Unimproved | 4.2 | 3.3 | 3.0 | 2.8 | 6,132 | 13.2 |
| Used a septic tank for stool | | | | | | |
| Yes | 25.5 | 17.8 | 12.5 | 9.6 | 30,257 | 65.3 |
| No | 10.7 | 8.9 | 8.0 | 7.1 | 16,058 | 34.7 |
| Used solid fuel | | | | | | |
| No | 27.0 | 18.7 | 13.0 | 10.2 | 31,920 | 68.9 |
| Yes | 9.2 | 8.0 | 7.5 | 6.5 | 14,395 | 31.1 |
| Paternal smoking | | | | | | |
| No | 13.6 | 9.3 | 6.7 | 5.5 | 16,227 | 35.0 |
| Yes | 22.6 | 17.4 | 13.7 | 11.3 | 30,088 | 65.0 |
| Wealth Quintile | | | | | | |
| Richest | 10.0 | 5.6 | 3.4 | 2.7 | 10,052 | 21.7 |
| Richer | 8.7 | 6.1 | 3.9 | 3.0 | 10,084 | 21.8 |
| Middle | 6.5 | 5.5 | 4.2 | 3.3 | 9,047 | 19.5 |
| Poorer | 5.8 | 5.0 | 4.3 | 3.5 | 8,616 | 18.6 |
| Poorest | 5.1 | 4.5 | 4.6 | 4.2 | 8,516 | 18.4 |
| Number of household members | | | | | | |
| < = 4 | 19.1 | 13.5 | 9.9 | 8.3 | 23,515 | 50.8 |
| >4 | 17.1 | 13.1 | 10.6 | 8.4 | 22,800 | 49.2 |
| Maternal education completed | | | | | | |
| High School | 16.3 | 10.0 | 6.5 | 5.2 | 17,589 | 38.0 |
| Secondary school | 7.7 | 6.3 | 5.0 | 3.9 | 10,632 | 23.0 |
| Primary school | 8.7 | 7.5 | 6.4 | 5.3 | 12,961 | 28.0 |
| No graduation | 3.4 | 2.7 | 2.6 | 2.3 | 5,133 | 11.1 |
| Paternal education completed | | | | | | |
| ≥ High School | 17.5 | 11.1 | 7.3 | 5.8 | 19,340 | 41.8 |
| Secondary school | 7.2 | 5.7 | 4.3 | 3.5 | 9,652 | 20.8 |
| Primary school | 8.2 | 7.0 | 6.2 | 5.1 | 12,273 | 26.5 |
| No graduation | 3.3 | 2.7 | 2.6 | 2.3 | 5,050 | 10.9 |
| Maternal occupation | | | | | | |

(*Continued*)

**Table 1.** (Continued)

| Risk factors | Normal | Mild stunting | Moderate stunting | Severe stunting | Total | |
|---|---|---|---|---|---|---|
| | % | % | % | % | n | % |
| Office employee | 4.6 | 2.3 | 1.5 | 1.2 | 4,458 | 9.6 |
| Entrepreneurs | 3.2 | 2.3 | 1.6 | 1.2 | 3,867 | 8.3 |
| Farmer | 4.4 | 3.7 | 3.5 | 3.4 | 6,958 | 15.0 |
| Low wages | 0.9 | 0.8 | 0.6 | 0.5 | 1,288 | 2.8 |
| Others | 1.5 | 1.2 | 1.0 | 0.6 | 2,021 | 4.4 |
| Unemployed | 21.5 | 16.3 | 12.2 | 9.8 | 27,723 | 59.9 |
| Paternal occupation | | | | | | |
| Office employee | 9.5 | 5.6 | 3.5 | 2.6 | 9,802 | 21.2 |
| Entrepreneurs | 9.5 | 6.5 | 4.6 | 3.7 | 11,281 | 24.4 |
| Farmer | 9.9 | 8.5 | 7.6 | 6.7 | 15,184 | 32.8 |
| Low wages | 4.5 | 3.8 | 3.1 | 2.3 | 6,342 | 13.7 |
| Others | 1.7 | 1.5 | 1.0 | 0.8 | 2,295 | 5.0 |
| Unemployed | 1.1 | 0.8 | 0.6 | 0.5 | 1,411 | 3.0 |
| Residence | | | | | | |
| Urban | 18.6 | 12.3 | 8.3 | 6.3 | 21,085 | 45.5 |
| Rural | 17.6 | 14.4 | 12.2 | 10.4 | 25,230 | 54.5 |

Notes: MUAC: Mid-upper arm circumference, BMI: Body mass index.

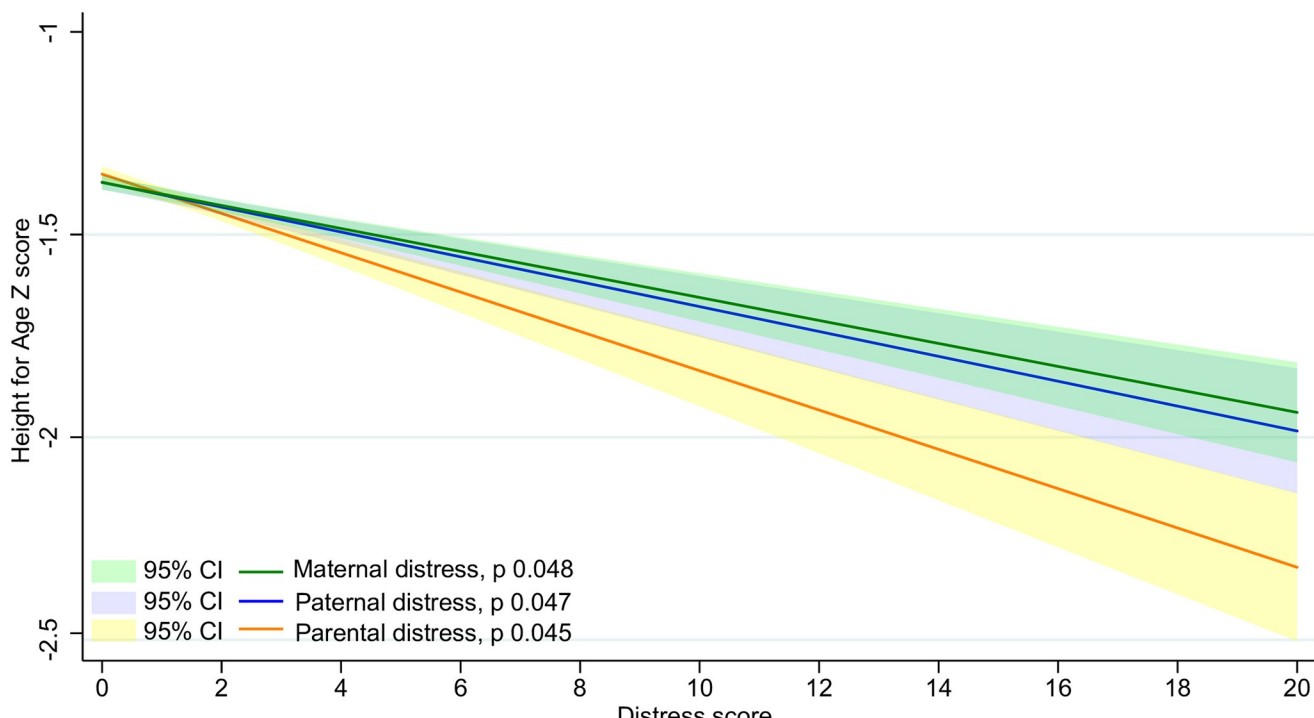

**Fig 3. The association between distress of parents and child height-for-age *z* score.** Distress is SRQ20 score, p values indicate significance of the regression coefficient for Height-for-Age Z score versus Distress score, shaded regions are the 98% CI for the fitted line.

**Table 2. Regression coefficient of distress of parents and other risk factors for child HAZ-score.**

| Risk factors | Full model | | | Best model | | |
|---|---|---|---|---|---|---|
| | β | 95% CI | p | β | 95% CI | p |
| Parental distress | | | | | | |
| No distress | 0 | | | 0 | | |
| Maternal distress | -0.089 | [-0.18,0.00] | 0.068 | -0.086* | [-0.17,-0.00] | 0.048 |
| Paternal distress | -0.082 | [-0.20,0.04] | 0.178 | -0.11* | [-0.21,-0.00] | 0.047 |
| Parental distress | -0.19 | [-0.40,0.02] | 0.070 | -0.19* | [-0.37,-0.01] | 0.045 |
| Had infectious diseases | | | | | | |
| No disease | 0 | | | 0 | | |
| ≥1 disease | -0.045* | [-0.08,-0.01] | 0.026 | -0.052** | [-0.09,-0.01] | 0.006 |
| Sex of the child | | | | | | |
| Girl | 0 | | | 0 | | |
| Boy | -0.051** | [-0.09,-0.02] | 0.006 | -0.065*** | [-0.10,-0.03] | 0.000 |
| Maternal BMI | | | | | | |
| Normal weight | 0 | | | | | |
| Underweight | -0.032 | [-0.11,0.05] | 0.428 | | | |
| Obese | 0.060 | [-0.00,0.12] | 0.059 | | | |
| Maternal MUAC (cm) | | | | | | |
| ≥ 23.5 | 0 | | | 0 | | |
| < 23.5 | -0.13*** | [-0.18,-0.07] | 0.000 | -0.14*** | [-0.19,-0.09] | 0.000 |
| Maternal height (cm) | | | | | | |
| ≥ 150 | 0 | | | 0 | | |
| < 150 | -0.43*** | [-0.47,-0.39] | 0.000 | -0.43*** | [-0.46,-0.39] | 0.000 |
| Paternal height (cm) | | | | | | |
| ≥ 155 | 0 | | | 0 | | |
| < 155 | -0.33*** | [-0.40,-0.26] | 0.000 | -0.31*** | [-0.38,-0.25] | 0.000 |
| Maternal age (years) | | | | | | |
| <25 | 0 | | | | | |
| 25–34 | 0.033 | [-0.03,0.10] | 0.320 | | | |
| ≥ 35 | 0.044 | [-0.03,0.12] | 0.270 | | | |
| Paternal age (years) | | | | | | |
| <30 | 0 | | | | | |
| 30–39 | -0.024 | [-0.08,0.03] | 0.418 | | | |
| ≥ 40 | 0.000087 | [-0.07,0.07] | 0.998 | | | |
| Iodized salt used | | | | | | |
| Yes | 0 | | | 0 | | |
| No | -0.073** | [-0.12,-0.02] | 0.004 | -0.075*** | [-0.12,-0.03] | 0.001 |
| Growth Monitoring attended (times) | | | | | | |
| = 6 | 0 | | | | | |
| < 6 | -0.022 | [-0.06,0.02] | 0.284 | | | |
| Used poor water quality | | | | | | |
| No | 0 | | | | | |
| Yes | 0.042 | [-0.04,0.12] | 0.304 | | | |
| Used unprocessed water | | | | | | |
| Yes | 0 | | | | | |
| No | -0.014 | [-0.06,0.03] | 0.545 | | | |
| Washed hand | | | | | | |
| Yes | 0 | | | | | |

(*Continued*)

**Table 2.**  (Continued)

| Risk factors | Full model | | | Best model | | |
|---|---|---|---|---|---|---|
| | β | 95% CI | p | β | 95% CI | p |
| No | -0.023 | [-0.06,0.02] | 0.268 | | | |
| Poor garbage disposal | | | | | | |
| No | 0 | | | 0 | | |
| Yes | -0.100*** | [-0.15,-0.05] | 0.000 | -0.097*** | [-0.14,-0.05] | 0.000 |
| Used of water waste disposal | | | | | | |
| Improved | 0 | | | | | |
| Unimproved | -0.043 | [-0.09,0.01] | 0.103 | | | |
| Used of toilet | | | | | | |
| Improved | 0 | | | | | |
| Unimproved | 0.027 | [-0.045,0.100] | 0.463 | | | |
| Used septic tank for stool | | | | | | |
| Yes | 0 | | | | | |
| No | -0.012 | [-0.072,0.047] | 0.688 | | | |
| Used solid fuel | | | | | | |
| No | 0 | | | | | |
| Yes | -0.010 | [-0.070,0.049] | 0.732 | | | |
| Paternal smoking | | | | | | |
| No | 0 | | | 0 | | |
| Yes | -0.060** | [-0.10,-0.02] | 0.002 | -0.060** | [-0.09,-0.02] | 0.001 |
| Wealth Quintile | | | | | | |
| Richest | 0 | | | 0 | | |
| Richer | -0.12*** | [-0.17,-0.06] | 0.000 | -0.12*** | [-0.17,-0.07] | 0.000 |
| Middle | -0.21*** | [-0.27,-0.15] | 0.000 | -0.22*** | [-0.27,-0.16] | 0.000 |
| Poorer | -0.23*** | [-0.31,-0.15] | 0.000 | -0.23*** | [-0.30,-0.17] | 0.000 |
| Poorest | -0.29*** | [-0.41,-0.18] | 0.000 | -0.34*** | [-0.40,-0.27] | 0.000 |
| Number house member | | | | | | |
| < = 4 | 0 | | | | | |
| >4 | -0.049* | [-0.09,-0.01] | 0.014 | | | |
| Maternal education | | | | | | |
| High School | 0 | | | 0 | | |
| Secondary school | -0.075** | [-0.13,-0.02] | 0.007 | -0.084*** | [-0.13,-0.04] | 0.001 |
| Primary school | -0.063* | [-0.12,-0.00] | 0.042 | -0.087*** | [-0.14,-0.04] | 0.001 |
| No graduation | -0.071 | [-0.16,0.02] | 0.115 | -0.081* | [-0.15,-0.01] | 0.016 |
| Maternal occupation | | | | | | |
| Office employee | 0 | | | 0 | | |
| Entrepreneurs | -0.17*** | [-0.25,-0.08] | 0.000 | -0.15*** | [-0.23,-0.07] | 0.000 |
| Farmer | -0.20*** | [-0.30,-0.11] | 0.000 | -0.18*** | [-0.26,-0.10] | 0.000 |
| Low wages | -0.25*** | [-0.38,-0.12] | 0.000 | -0.21*** | [-0.33,-0.09] | 0.001 |
| Others | -0.14* | [-0.24,-0.03] | 0.011 | -0.12* | [-0.22,-0.02] | 0.021 |
| Unemployed | -0.13*** | [-0.19,-0.06] | 0.000 | -0.13*** | [-0.19,-0.06] | 0.000 |
| Paternal education | | | | | | |
| ≥ High School | 0 | | | | | |
| Secondary school | 0.0044 | [-0.05,0.06] | 0.875 | | | |
| Primary school | -0.025 | [-0.09,0.04] | 0.421 | | | |
| No graduation | -0.0092 | [-0.10,0.08] | 0.836 | | | |
| Paternal occupation | | | | | | |

(*Continued*)

**Table 2.** (Continued)

| Risk factors | Full model | | | Best model | | |
|---|---|---|---|---|---|---|
| | β | 95% CI | p | β | 95% CI | p |
| Office employee | 0 | | | 0 | | |
| Entrepreneurs | -0.027 | [-0.08,0.03] | 0.347 | -0.029 | [-0.08,0.02] | 0.279 |
| Farmer | -0.090** | [-0.16,-0.02] | 0.009 | -0.078* | [-0.14,-0.02] | 0.013 |
| Low wages | -0.077* | [-0.15,-0.01] | 0.033 | -0.070* | [-0.14,-0.00] | 0.037 |
| Others | -0.073 | [-0.16,0.02] | 0.122 | -0.094* | [-0.18,-0.01] | 0.034 |
| Unemployed | 0.089 | [-0.02,0.20] | 0.123 | 0.057 | [-0.05,0.16] | 0.285 |
| Residence | | | | | | |
| Urban | 0 | | | 0 | | |
| Rural | -0.052* | [-0.10,-0.00] | 0.037 | -0.062** | [-0.11,-0.02] | 0.007 |

1.19; p = 0.023) stunting, but not severe stunting. Children who had fathers with smoking behavior had a higher risk for moderate (RR = 1.07, 95%CI 1.01 to 1.13; p = 0.016) and severe (RR = 1.08, 95%CI 1.02 to 1.15; p = 0.002) stunting.

In addition, socioeconomic factors related to stunting were household wealth quintile, number of household members, maternal education, maternal and paternal occupation, and rural or urban residence. The analysis showed that quintiles of lower wealth were associated with mild, moderate, and severe stunting. Children from the poorest quintile had a 1.77-fold

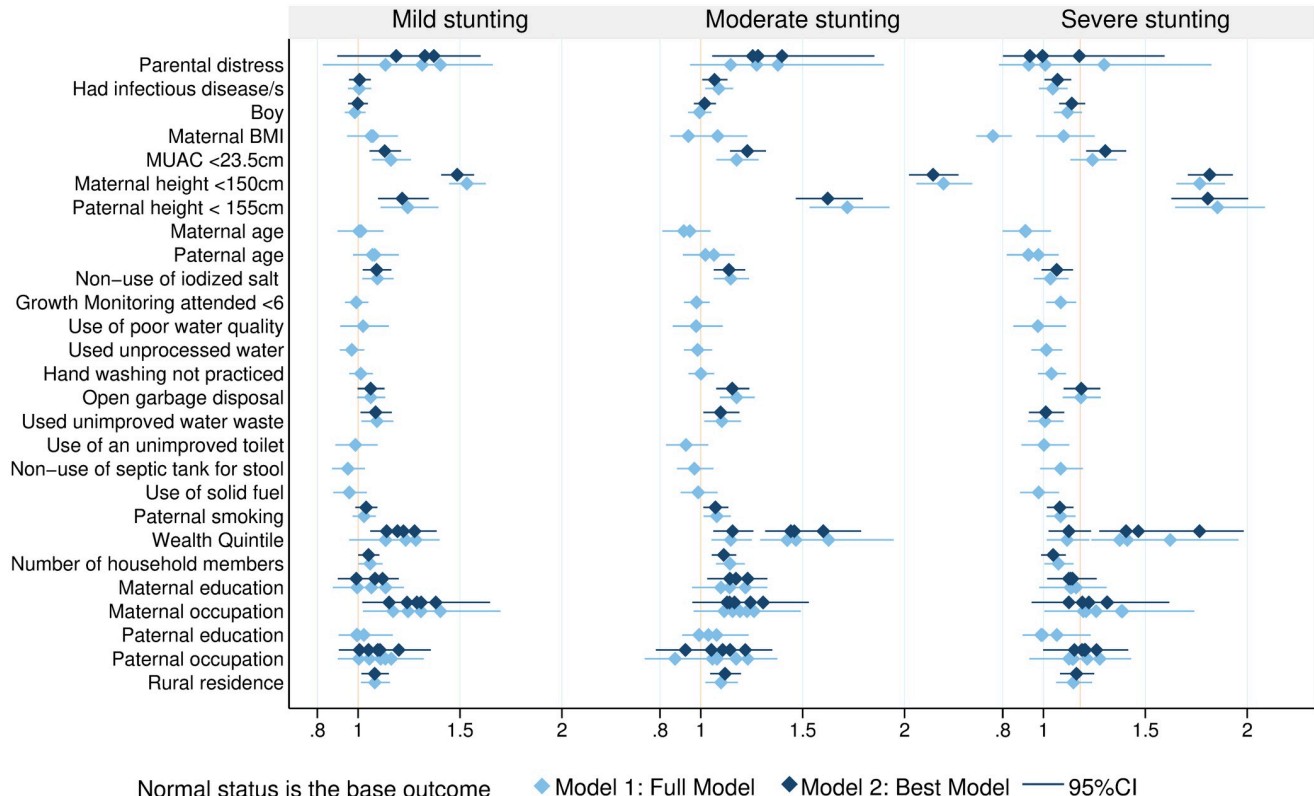

**Fig 4. The relative risk ratio of distress of parents and other factors for stunting of children age 6–59 months.** Multiple point estimates shown for some categories indicate the risks and 95% confidence intervals (CI) for the specific subcategories in comparison to the reference group, as shown in Table 2.

higher risk of severe stunting (95%CI 1.57 to 1.98; p< 0.001). The risk gradually reduced from moderately poor (RR = 1.47, 95%CI 1.32 to 1.63; p<0.001), middle (RR = 1.41, 95%CI 1.27 to 1.55; p<0.001) to richest quintile (1.12, 95%CI 1.03 to 1.23; p = 0.013). Lower maternal education levels, not completing high school compared to completing high school, were associated with higher risk of mild (RR = 1.12, 95%CI 1.05 to 1.20; p = 0.001), moderate (RR = 1.23, 95% CI 1.14 to 1.33; p<0.001) and severe (RR = 1.14, 95%CI 1.05 to 1.24; p = 0.001) stunting. Maternal and paternal occupations were significantly associated with stunting. Compared with children of office employee mothers or fathers, children whose mothers or fathers worked as entrepreneurs, farmers, low wage employees, or were unemployed were more at risk for mild, moderate, and severe stunting. As entrepreneurs, mothers had 1.29 (95%CI 1.14 to 1.45; p = <0.001), 1.24 (95%CI 1.09 to1.42; p = 0.001), and 1.22 (95%CI 1.06 to 1.42; p = 0.007) times higher risk to have mild, moderate, and severely stunted children. Children living in rural areas had 1.08 (95%CI 1.02 to 1.15; p = 0.013), 1.12 (95%CI 1.05 to 1.20; p = 0.001), and 1.16 (95%CI 1.08 to 1.25; p<0.001) higher risk of stunting compared to children living in urban areas.

To further explore how parental distress might affect stunting, we assessed the association of parental distress with several factors associated with child stunting, including infectious disease episodes, non-use used of iodized salt, use of solid fuel, paternal smoking, use of open garbage and unimproved water disposal, and poor hand-washing behavior (Table 3).

The adjusted relative risk ratios of maternal, paternal, and parental distress for children who experienced at least one infectious disease were 1.59 (95%CI 1.43 to 1.77, p<0.001), 1.26 (95%CI 1.10 to 1.44, p = 0.001), and 1.78 (95%CI 1.42 to 2.22, p<0.001). Children of families with higher maternal, paternal, and parental distress were at higher risk for not using iodized salt with adjusted relative risk ratios of 1.23 (95%CI 1.05 to 1.44, p = 0.009), 1.22(95%CI 1.00 to 1.48, p = 0.049), 1.52 (95%CI 1.10 to 2.09, p = 0.010). Furthermore, maternal, paternal, and parental distress were associated with the use of poor water quality, poor handwashing behavior, poor garbage disposal, and paternal smoking. However, parental distress was not associated with attendance to growth monitoring sessions, use of unprocessed water, use of a septic tank, use of unimproved water waste disposal, use of an unimproved toilet, non-use of a septic tank, nor use of solid fuel for cooking.

Due to the association of multiple risk factors with stunting, and the association of distress with multiple risk factors, we quantified the proportion of HAZ-score lost in the population using the regression coefficient beta values from Table 2 (presented in Table S2 Appendix) and plotted these against the adjusted relative risk ratio of parental distress associated with these same factors (presented in Table S3 Appendix). Results are shown in Fig 5.

Amongst the specific risk factors, more than 50% of HAZ-score lost was due to low wealth (12.3%), short maternal stature (9.9%), low maternal occupation (9.0%), poor garbage disposal (5.0%), low maternal education (3.7%), paternal smoking (2.8%), low paternal occupation (2.8%), rural residence (2.4%), and being a boy child (2.3%). For specific groups of risk factors, the highest proportion of HAZ-score lost was 30.3% associated with socioeconomic factors, including low household wealth, low maternal occupation status, low maternal education, rural residence and low paternal occupation status. The second group was physiological factors, associated with a HAZ-score loss of 15.5% including maternal short stature, being a boy child, paternal short stature, and low maternal MUAC followed by behavioral factors with 8.9% including used of poor garbage sanitation, paternal smoking, and did not using iodized salt. Parental distress as an isolated risk factor accounted for 0.6% of HAZ-score loss.

Notably, parental stress was associated with increased risk by at least 1.5-fold for the three largest contributors to HAZ-lost, specifically, low wealth, short maternal stature, and low maternal occupation. We calculated that 96.4% of children reside in a household with any of

**Table 3. Parental distress relation to behavioral and morbidity risk factors of stunting.**

| | Not used Iodized salt | | | Low attendance of growth monitoring <6 | | | Used poor quality water | | |
|---|---|---|---|---|---|---|---|---|---|
| | RR | 95% CI | p | RR | 95% CI | p | RR | 95% CI | p |
| *Unadjusted* | | | | | | | | | |
| No distress | 1 | | | 1 | | | 1 | | |
| Maternal distress | 1.35*** | [1.16,1.58] | <0.001 | 0.95 | [0.85,1.07] | 0.409 | 1.67*** | [1.36,2.06] | <0.001 |
| Paternal distress | 1.33** | [1.10,1.61] | 0.004 | 0.99 | [0.86,1.14] | 0.839 | 1.46** | [1.12,1.90] | 0.005 |
| Parental distress | 1.77*** | [1.29,2.44] | <0.001 | 1.02 | [0.80,1.29] | 0.880 | 2.40*** | [1.63,3.55] | <0.001 |
| *Adjusted^* | | | | | | | | | |
| No distress | 1 | | | 1 | | | 1 | | |
| Maternal distress | 1.23** | [1.05,1.44] | 0.009 | 0.99 | [0.88,1.11] | 0.837 | 1.52*** | [1.24,1.88] | <0.001 |
| Paternal distress | 1.22* | [1.00,1.48] | 0.049 | 1.03 | [0.89,1.18] | 0.709 | 1.31* | [1.01,1.70] | 0.045 |
| Parental distress | 1.52* | [1.10,2.09] | 0.010 | 1.10 | [0.86,1.40] | 0.439 | 1.97*** | [1.33,2.91] | 0.001 |
| | **Used unprocessed water** | | | **No hand washing** | | | **Poor garbage disposal** | | |
| *Unadjusted* | | | | | | | | | |
| No distress | 1 | | | 1 | | | 1 | | |
| Maternal distress | 0.94 | [0.82,1.09] | 0.417 | 1.30*** | [1.15,1.47] | <0.001 | 1.00 | [0.84,1.19] | 0.994 |
| Paternal distress | 0.91 | [0.77,1.09] | 0.310 | 1.19* | [1.02,1.38] | 0.026 | 1.07 | [0.86,1.33] | 0.532 |
| Parental distress | 0.74 | [0.54,1.01] | 0.054 | 1.58*** | [1.23,2.03] | <0.001 | 1.74** | [1.17,2.59] | 0.006 |
| *Adjusted^* | | | | | | | | | |
| No distress | 1 | | | 1 | | | 1 | | |
| Maternal distress | 1.11 | [0.96,1.28] | 0.176 | 1.20** | [1.07,1.36] | 0.003 | 0.83* | [0.70,0.99] | 0.038 |
| Paternal distress | 1.08 | [0.90,1.29] | 0.421 | 1.09 | [0.94,1.27] | 0.254 | 0.87 | [0.70,1.08] | 0.212 |
| Parental distress | 0.97 | [0.70,1.33] | 0.829 | 1.38* | [1.08,1.77] | 0.011 | 1.27 | [0.86,1.87] | 0.238 |
| | **Used unimproved water waste** | | | **Used unimproved toilet** | | | **No used septic tank for stool** | | |
| *Unadjusted* | | | | | | | | | |
| No distress | 1 | | | 1 | | | 1 | | |
| Maternal distress | 1.31** | [1.09,1.58] | 0.004 | 1.26* | [1.03,1.55] | 0.025 | 1.42*** | [1.23,1.64] | <0.001 |
| Paternal distress | 1.34* | [1.06,1.69] | 0.014 | 1.40** | [1.09,1.80] | 0.008 | 1.41*** | [1.19,1.67] | <0.001 |
| Parental distress | 1.10 | [0.75,1.62] | 0.621 | 1.32 | [0.86,2.01] | 0.201 | 1.67*** | [1.25,2.24] | 0.001 |
| *Adjusted^* | | | | | | | | | |
| No distress | 1 | | | 1 | | | 1 | | |
| Maternal distress | 1.16 | [0.96,1.40] | 0.129 | 0.84 | [0.65,1.08] | 0.174 | 1.03 | [0.86,1.24] | 0.748 |
| Paternal distress | 1.12 | [0.88,1.42] | 0.343 | 1.09 | [0.80,1.47] | 0.593 | 1.06 | [0.85,1.33] | 0.601 |
| Parental distress | 0.84 | [0.57,1.26] | 0.404 | 0.74 | [0.44,1.25] | 0.266 | 1.02 | [0.69,1.50] | 0.919 |
| | **Used solid fuel** | | | **Paternal smoking** | | | **Experienced infectious disease** | | |
| *Unadjusted* | | | | | | | | | |
| No distress | 1 | | | 1 | | | 1 | | |
| Maternal distress | 1.35*** | [1.15,1.58] | <0.001 | 1.32*** | [1.19,1.47] | <0.001 | 1.61*** | [1.45,1.80] | <0.001 |
| Paternal distress | 1.40*** | [1.16,1.69] | <0.001 | 1.26*** | [1.10,1.44] | 0.001 | 1.29*** | [1.12,1.47] | <0.001 |
| Parental distress | 2.35*** | [1.71,3.25] | <0.001 | 1.67*** | [1.31,2.11] | <0.001 | 1.81*** | [1.45,2.26] | <0.001 |
| *Adjusted^* | | | | | | | | | |
| No distress | 1 | | | 1 | | | 1 | | |
| Maternal distress | 0.96 | [0.81,1.15] | 0.682 | 1.23*** | [1.10,1.37] | <0.001 | 1.59*** | [1.43,1.77] | <0.001 |
| Paternal distress | 0.97 | [0.78,1.20] | 0.754 | 1.15* | [1.00,1.31] | 0.043 | 1.26*** | [1.10,1.44] | 0.001 |
| Parental distress | 1.37 | [0.96,1.95] | 0.079 | 1.43** | [1.13,1.82] | 0.003 | 1.78*** | [1.42,2.22] | <0.001 |

RR = Risk Ratio using logistic regression,

^adjusted by maternal and paternal age, education, occupation, and wealth.

* $p < 0.05$,

** $p < 0.01$,

*** $p < 0.001$.

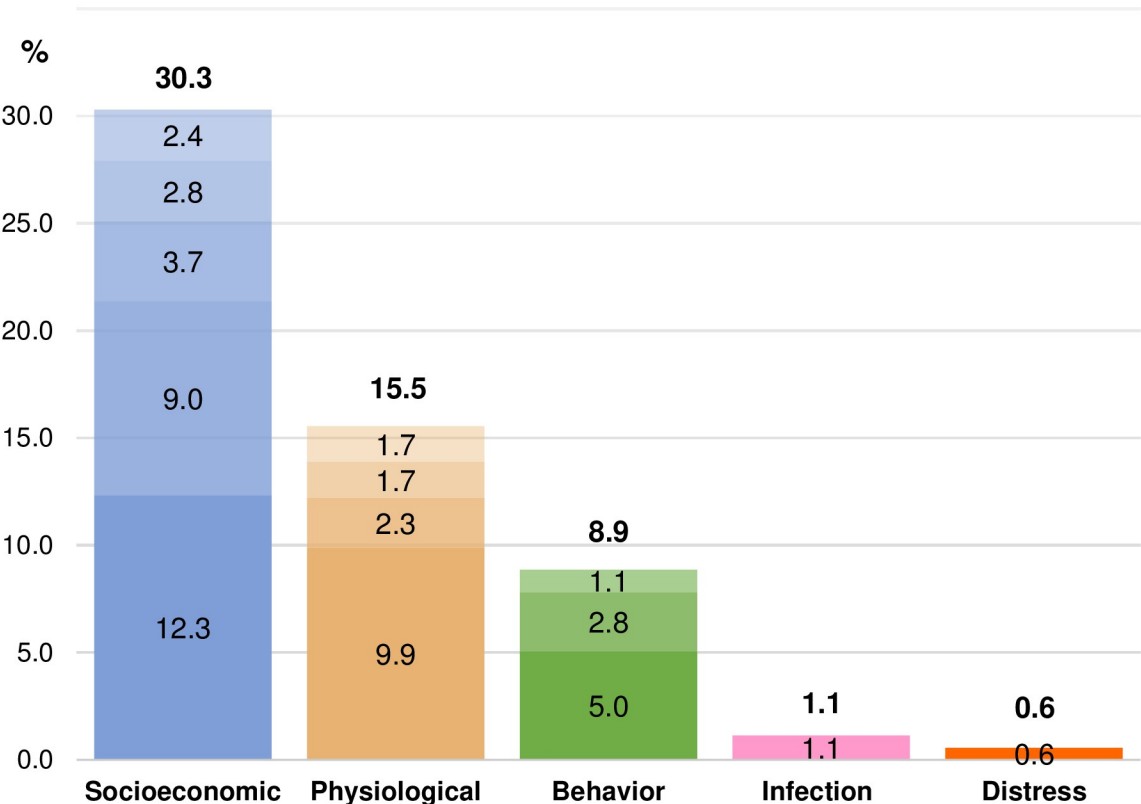

Socioeconomic factors (from lowest stack) account for 30.3% of z-score lost, include low wealth, low maternal occupational status, low maternal education, rural residence, low paternal occupational status.

Physiological factors account for 15.5% include maternal height < 150cm, boys, paternal height < 155 cm, maternal MUAC< 23.5 cm.

Health behavior factors account for 8.9% include poor garbage disposal, paternal smoking, not using iodized salt.

Infectious disease factors account for 1.1%, experienced with one or more of infectious diseases.

Parental distress account for 0.6 %

**Fig 5. The proportion of HAZ-score lost associated with parental distress and other specific factors.**

these risks. Parental stress also increased the associated risk more than 2-fold for three of the top nine contributors to HAZ-lost, specifically, low maternal and paternal occupation status, and low maternal education, which could affect 92.0% of children. These groups may therefore represent substantial proportions of the populations wherein enhanced parental stress may be particularly problematic.

## Discussion

In this study, we found that maternal, paternal, and parental distress are associated with reduced linear growth of children. For each one-unit increase in maternal, paternal and parental distress, the HAZ -score of the child decreased by 0.09, 0.11 and 0.19, respectively. We were unable to find a previous study that yielded the relationship between parental and paternal distress on stunting or growth of children. Paternal distress has been commonly known to relate to low income and poverty, job insecurity, and social inequalities [39]. We also note the

observed effect of maternal distress is larger than previously reported by other studies in rural Bangladesh where infants at age 12 months whose mothers had symptoms of depression had a 0.007 drop in mean HAZ [40]. Our finding is in agreement with a longitudinal study of a nationally representative US birth cohort showed mothers with mild and moderate depressive symptoms had children who were, on average, 0.26 cm shorter in stature (95% CI: -0.48 cm to -0.05 cm) than their peers [41].

Our analyses indicate that parental distress is related to mild and moderate stunting, but not severe stunting. A previous study suggested that maternal stress and/or depression may affect disturbance of the hypothalamic-pituitary-adrenal (HPA) axis thereby exerting physiological effects during intrauterine growth [42]. This mechanism affects fetal weight which is mediated by prenatal cortisol and norepinephrine levels. Maternal distress also affects inadequate nutritional care. A depressed mother may experience fatigue, impaired concentration, and psychomotor slowing, all of which could affect feeding practices. The relative risk ratios of maternal, paternal, and parental distress for mild stunting ranged from 1.23 to 1.35 and for moderate stunting from 1.25 to 1.44. This was slightly higher than reported for cohort studies in India and Peru with odds ratios of 1.18 (95% CI 1.03 to 1.35) and 1.24 (95% CI 1.07 to 1.44), respectively [43]. But the results are more similar to an Indian study of maternal mental health on child stunting at 6–18 months with OR 1.4 (95%CI 1.2. to 1.6) [44], and a study from South Africa reporting an OR 1.61 (95% CI 1.02 to 2.56) [45] for stunting of children aged 2 years if their mother was socioeconomically disadvantaged and depressed (24). A higher OR of 2.17 was reported for infants of mothers with depressive symptoms in rural Bangladesh [45]. A meta-analysis found an OR was 1.4 (95%CI 1.2. to 1.7) [46]. As such, several studies support the relation of maternal mental distress or disorder on linear growth.

As mentioned, associations of paternal distress and parental distress suffered by both parents on stunting and linear growth have not been previously reported. Paternal and parental distress showed a similar association with linear growth of their children as compared to maternal distress. The generation R study conducted in the high-income country of the Netherlands, which reported maternal psychological distress was positively associated with overweight children, found inconsistent associations between paternal distress and child anthropometry [47].

The present study found that distress of parents was associated with reduced linear growth of their children. More than half of under-five children were with mild, moderate, or severe stunting. Based on the population projection of Indonesia in 2013 there were approximately 23,994,200 under-five children [48], this would account for 15.31 million stunted children. And 7.5% of under-five children experienced parental distress—either maternal (4.1%) or paternal (2.6%) or both (0.9%). Of those children exposed to any type of distress of parents, approximately two-thirds (70.4%) had mild, moderate, or severe stunting. Based on these data we estimate approximately 1.27 million stunted children in Indonesia reside in households experiencing parental distress. This is a substantial burden that is likely to be underestimated. We note the proportion of households with psychological distress from the SRQ20 at the provincial level ranged from 1.6% to 15.9%, with urban centers tending toward higher stress levels, suggesting this burden is likely to increase as urbanization is increasing globally, and under acute stress conditions such as the Covid-19 pandemic. Under-five children are at a critical stage of development, needing stimulation and attention [49,50]. Distress among parents may affect parenting quality, especially responsiveness to child interactions, or exacerbate toxic stress that is known to affect child growth and development [51,52].

We found that distress of parents was associated with multiple risk factors for stunting, suggesting a pleiotropic effect of distress. Parental distress may interfere with caregiving behavior which may lead to poor child growth and morbidity [53,54]. Parents experiencing distress

tend to be less engaged with their children's daily activities and pay less attention to their growth, which also implicates less responsive care and feeding practice. Parents may fail to acknowledge the child's cues of hunger, which can lead to poor nutrition. Also, parental distress may influence children's stress levels, which may also influence their growth [41,55]. Another mechanism that may increase the risk of both parental distress and stunting is poverty. Our analysis showed that socioeconomic status was linearly related to child stunting. Poverty is the source of many stunting risk factors including inadequate health care and lack of food security [53].

In our study parental distress was also associated with paternal smoking, exposure to unimproved waste and water disposal, use of unimproved water supplies, having a septic tank for stools, non-iodized salt use, and poor hand-washing behavior. Since parenting was included in the UNICEF Framework for Improved Nutrition of Children and Women in Developing Countries in 1990 [56], there have been relatively few efforts to address this problem. This study suggests that the promotion of parenting at the household level may be important for the betterment of child growth. Whereas most modifiable factors affecting undernutrition and stunting could be addressed by parenting at the household level, distressed parents might not be able to cope as well with the required actions. We recommend the promotion of mental and behavioral health for parents, which would promote child growth and also promote early childhood cognitive development. The actions taken for mental health could be for developing and protecting individual attributes, supporting households and communities, and supporting vulnerable groups in society [39]. The actions for behavioral health promotion for the parent could be integrated with nutrition-specific and sensitive activities in stunting reduction programs [57].

We note that while parental distress directly accounted for only 0.6% of the overall loss of HAZ-score, its indirect effects are substantial, with strong effects on risks affecting nearly 96% of the population. Based on previous studies, this would extend to a risk factors for poor parenting [58,59], food security and consumption [22,60], morbidity [54], and health care access [61]. Also, children must live in a safe, supportive, and nurturing family, and parental distress could interfere with the rights of the child [62]. Other findings from this study are noteworthy. Children suffering from even one infectious disease (diarrhea, upper respiratory tract infection, malaria) had an elevated risk for moderate and severe stunting. Physiological factors that relate to child stunting were MUAC, maternal and paternal height, and child sex. For behavioral factors, several were found to be significant determinants of stunting. Specifically, use of an open garbage disposal, unimproved water waste disposal, paternal smoking, and not using iodized salt were all important. The socioeconomic factors correlated with child stunting were household wealth, maternal education, mothers' and fathers' occupation, and rural residence. These have been reported by other studies [5,63–67]. Behavior modification interventions should improve child growth by targeting improved garbage disposal, water waste disposal, and smoking cessation. Intervention on social conditions should include increasing household income, improving maternal education, providing jobs, and attention to rural mothers' conditions and working mothers. These interventions could also support reduced prevalence of distress [39].

Our analysis assessed maternal, paternal, and parental distress on linear growth adjusted for multiple risk factors not previously examined as a group. Paternal and parental distress showed a similar association with linear growth as compared to maternal distress. To our knowledge, our study represents the most comprehensive analysis examining the association between parental distress from a nationally representative sample involving more than 46,000 two-parent families.

However, this study has limitations. First, this was a cross-sectional study that could not formally infer causality. Second, this study did not include one-parent families, which may be a considerable proportion of households with negative mental health consequences [68]. Third, some factors were not included in this study as predictors of linear growth including nutrition intake, child weight or length at birth, breastfeeding, food/nutrient supplementations, and feeding behavior. However, because many of these, such as birth weight and length are strongly correlated with included predictors such as maternal MUAC, maternal education, and maternal height, the findings regarding contributions of parental distress are likely to be robust.

Concerns regarding maternal distress in relation to child growth have been rising, and a better understanding is needed of interventions for maternal, paternal, and parental distress. This would include more studies to understand how maternal, paternal, and parental distress could reduce the linear growth of children. It could activate stress response through the HPA axis to change intestinal permeability, motility, and mucus production. The HPA axis activation also affects immune cell responses [69]. If the findings herein are confirmed, promotion of mental and behavioral health programs must be pursued as part of a comprehensive strategy to enhance child growth and development that can improve caretaker capacity and parenting skills, reduce gender discrimination, reduce domestic violence, and, integrate with community development.

## Conclusions

This analysis of nationally representative data found that distress of parents was associated with reduced linear growth and the risk of mild and moderate stunting of children. Parental distress was found to have strong associations with behavioral and other risk factors that limit child growth. As such, while the cumulative loss of height contributed from parental distress itself was limited, the potential effects of parental distress on overall linear growth via indirect effects are substantial. Given that multiple behavioral and socioeconomic factors influence stunting, we suggest promotion of mental and behavioral health programs for parents must be pursued as part of a comprehensive strategy to enhance child growth and development, improve caretaker capacity and parenting skills, reduce gender discrimination, reduce domestic violence, and integrate with community development.

## Supporting information

**S1 Appendix. Relative risk ratio of parental distress and other risk factors associated with stunting.**
(DOCX)

**S2 Appendix. HAZ-score loss associated with distress of parents and other risk factors.**
(DOCX)

**S3 Appendix. Association of distress of parents with other risk factors for stunting.**
(DOCX)

## Acknowledgments

The authors are grateful to the thousands of participants who provided all the information, and to the numerous person who contributed to the data collection and management, and to NIHRD who allowed us to use the dataset. KAS would like to thank the Higher Education Network Ring Initiative (HENRI) for allowing her to participate in the course and workshop by

AHS at the Harvard T.H. Chan School of Public Health entitled 'Analysis of Health and Nutrition Data from Low and Middle-Income Countries'. This paper was the output of that course.

## Author Contributions

**Conceptualization:** Kun A. Susiloretni, Emily R. Smith, Anuraj H. Shankar.

**Data curation:** Kun A. Susiloretni, Suparmi.

**Formal analysis:** Kun A. Susiloretni, Emily R. Smith, Suparmi, Anuraj H. Shankar.

**Investigation:** Kun A. Susiloretni, Marsum, Rina Agustina, Anuraj H. Shankar.

**Methodology:** Kun A. Susiloretni, Emily R. Smith, Rina Agustina, Anuraj H. Shankar.

**Software:** Kun A. Susiloretni, Suparmi.

**Supervision:** Anuraj H. Shankar.

**Validation:** Suparmi, Marsum, Anuraj H. Shankar.

**Visualization:** Kun A. Susiloretni, Emily R. Smith, Marsum, Rina Agustina, Anuraj H. Shankar.

**Writing – original draft:** Kun A. Susiloretni, Suparmi, Anuraj H. Shankar.

**Writing – review & editing:** Kun A. Susiloretni, Emily R. Smith, Suparmi, Marsum, Rina Agustina, Anuraj H. Shankar.

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
