## [Decision Letter · Decision Letter 0]

21 Apr 2020

PONE-D-20-08335

The psychological distress of parents is associated with reduced linear growth of children: evidence from a nationwide population survey

PLOS ONE

Dear Dr. Susiloretni,

Thank you for submitting your manuscript to PLOS ONE. After careful consideration, we feel that it has merit but does not fully meet PLOS ONE’s publication criteria as it currently stands. Therefore, we invite you to submit a revised version of the manuscript that addresses the points raised during the review process.

Please address all reviewers´comments in the revised version of the manuscript. 

We would appreciate receiving your revised manuscript by Jun 05 2020 11:59PM. To enhance the reproducibility of your results, we recommend that if applicable you deposit your laboratory protocols in protocols.io, where a protocol can be assigned its own identifier (DOI) such that it can be cited independently in the future. For instructions see: http://journals.plos.org/plosone/s/submission-guidelines#loc-laboratory-protocols

We look forward to receiving your revised manuscript.

Kind regards,

Marly A. Cardoso, Ph.D.

Academic Editor

PLOS ONE

Journal Requirements:

1) Please provide additional details regarding participant consent. In the ethics statement in the Methods and online submission information, please ensure that you have specified whether consent was informed.

2) Please improving statistical reporting and refer to p-values as "p<.001" instead of "p=.000". Our statistical reporting guidelines are available at https://journals.plos.org/plosone/s/submission-guidelines#loc-statistical-reporting

3) Please do not include funding sources in the Acknowledgments or anywhere else in the manuscript file. Funding information should only be entered in the financial disclosure section of the submission system. https://journals.plos.org/plosone/s/submission-guidelines#loc-acknowledgments

4) We note that you have stated that you will provide repository information for your data at acceptance. Should your manuscript be accepted for publication, we will hold it until you provide the relevant accession numbers or DOIs necessary to access your data. If you wish to make changes to your Data Availability statement, please describe these changes in your cover letter and we will update your Data Availability statement to reflect the information you provide.

5) Please include captions for your Supporting Information files at the end of your manuscript, and update any in-text citations to match accordingly. Please see our Supporting Information guidelines for more information: http://journals.plos.org/plosone/s/supporting-information

Reviewers' comments:

Reviewer's Responses to Questions

**Comments to the Author**

1. Is the manuscript technically sound, and do the data support the conclusions?

Reviewer #1: Yes

Reviewer #2: Partly

2. Has the statistical analysis been performed appropriately and rigorously? 

Reviewer #1: Yes

Reviewer #2: Yes

3. Have the authors made all data underlying the findings in their manuscript fully available?

Reviewer #1: Yes

Reviewer #2: No

4. Is the manuscript presented in an intelligible fashion and written in standard English?

Reviewer #1: No

Reviewer #2: Yes

5. Review Comments to the Author

Reviewer #1: The present study has important findings and highlights the need of a comprehensive attention to the child development. The findings of this study provide a good description of the studied population and present a range of “not obvious” factors that may have an essential role to an adequate child development. Additionally, the discussion promotes a good dialogue between the findings of this study and the literature.

The introduction brings a great set of information about the topic and provides a contextualization of the subject to the reader, but also signalizes a gap in the literature about the theme, highlighting the potentialities and limitations of previous studies. However, in this section, there are some information without scientific references. For example, in the beginning of the manuscript, long- and short-term effects of the linear growth restriction (line 36-39) were point out, but there are no references. Next, the text quotes “multiple studies (…)” (line 41-43) and a “recent study” (line 43), but don’t provide the references of these studies.

Still in the introduction, there are some statements that are confusing, may leading to misinterpretation. For example, in the phrase “Additional risk in some studies include environmental factors, maternal health status, and child health status” (line 45), it is not clear what is in risk and how these factors where explored in each article.

The methodology section evidences the power of this study, proportionating good perspective of the study design to the reader and evidencing the large data set used. Nevertheless, some aspects should be reviewed. In the variables section, you stablish the cut off points to classify stunting as mild, moderate and severe (line 95-96), but it would be important to reference it (i.e. WHO child growth standards). Also, the creation process of the distress variables, using different score cut-off points, is not clear in the text (line 101-108). It was only possibly to fully understand the categorization of the variables when observing the tables.

Another aspect to be reviewed is the nature of the variables. You describe MUAC, BMI, height and age of the mother/father and child age as physiological factors (line 112-114), but these variables should be placed in distinct groups since their nature is different. It is important to note that physiological conditions are related to organic and biologic aspects of the human body. Next, you describe iodized salt, drinking water source, garbage disposal, water waste disposal, stool disposal as health behaviours (line 114-115). However, behaviours are more connected to individual actions then sanitation conditions and health access.

Still in the methodology, you describe the formula used to obtain the proportion of z-score lost due to any parental stress. However, the text is a bit confusing and not clear, making difficult to understand the construction this formula (line 128-132).

The results from the study are described in a very fine way. However, there are some important considerations in order to contribute to the quality of this section. In this segment you affirm that maternal, paternal and parental distress are associated with lower HAZ according to the figure 3 (line 153-154). Even though this figure brings information in a really clear way, there is no data about the p value in this figure.

Next, you say that parental distress accounted for a reduction of 0.36 HAZ (line 156-157), but this information is not available in the table 2.

Still in the results section, you say that paternal hight was associated with mild, moderate and severe, but don’t mention clearly if you are referring to stunting (line 188).

The discussion of the findings highlights important previous findings, intertwining the existing literature and the findings from the presented study. Yet, considerations must be taken in account to improve this section. In the beginning of the discussion you state that parental distress was associated with linear growth of their children. However, this is a cross sectional study, making it not possible to track linear growth. Also, there is an extensive description of the results in the first paragraph, not fitting into the discussion section requirements. On the other hand, some studies mentioned are not well described, making the dialogue between previous findings and the results not comparable (i.e. references number 32 and 34, 36, 37, 4, 40-44 not indicating the age of the studied population or methods). There is also information without references (line 302-305 and line 349-352).

The limitations and potentialities of the study are mentioned in the manuscript. However, it is important to highlight that this study did not include the one-parent families, which may be a considerable part of the initial households and may present different results.

An important caveat to point about the text concerns to the morbidity variable. When first describing the morbidity variable in the methodology, you say that “upper respiratory infection” was considered a disease (line 111), but latter you replace it with “acute respiratory infection” (line 177), which is not a synonym, but a different condition. In addition, in the results section you affirm that infectious disease was related to stunting, but then you include a condition that can be originated by an infection or not (ie. diarrhea) in this variable (line 177-178). Also, in table 2 there is only the variable “disease”, not presenting different results to each one of the nature of the disease (non-infectious versus infectious disease).

Overall, the manuscript is well structured and organized. Yet, the language quality should be verified, since many parts demands extra attention to interpret the information. Also, the language may compromise the understanding of the content, leading to ambiguity and doubts.

Reviewer #2: The psychological distress of parents is associated with reduced linear growth of children: evidence from a nationwide population survey

Review PONE-D-20-08335

The article aims "to identify whether the psychological distress of parents is related to child linear growth and stunting, and to document the associated risk factors, and examine the relationship between parental distress and other behavioral risk factors for stunting", for which analyzed secondary data (n=54,261 households) from the Indonesia National Health Survey 2013.

This is an interesting topic for public health, although the demonstration of the relationship between mental health and nutritional status of children is not new [1-7]. As strengths of the study could be cited: (a) the use of a nationally representative sample involving more than 50,000 families, and (b) the indication that parental distress, interacting with multiple other factors, accounted for 5.6% of the overall loss of HAZ-score. The main negative points were: (a) use of outcome classification criteria favoring sensitivity, at the expense of specificity, which generates many cases of false positives, weakening the strength of the evidence by overestimating the findings; (b) Overvaluing the results found in the unadjusted analysis, when it should focus on data from the adjusted analysis, when, in theory, the effects of confounding factors were reduced. In this aspect, numerous studies indicate that mental health disorders are associated with several risk factors common to undernutrition. Thus, the importance of this study would be to separate the specific contribution of this outcome in determining child growth deficit; (c) In order to value the theme explored (psychological distress of parents in the etiology of stunting), the authors minimize the role of socio-economic-demographic aspects. It is not true that "Although the etiology is recognized to be complex, most analyses have focused on biomedical determinants, with limited attention to social factors affecting care and nurturing in the home". In the old and well-known model for determining undernutrition, published by UNICEF [8], structural causes, such as political, economic and ideological structure, are recognized as basal (distal) determinants, preceding underlying causes and immediate causes. There is consensus that any interventions to address the problem of undernutrition in a population will not be sustainable if constant and permanent access to education, health care, adequate sanitation conditions and healthy food is not promoted.

More specifically, we present the following comments:

Lines 4 to 6: "Although the etiology is recognized to be complex, most analyses have focused on biomedical determinants, with limited attention to social factors affecting care and nurturing in the home". See comment above.

Lines 18: "paternal distress increased the risk of mild stunting (HAZ <-1) by 38% (95% CI 1.21,1.57)"… The proportion of children below - 1 sd in the healthy population (anthropometric pattern) is over 33%. Is it prudent to consider these children as having stunting? The rate of false positives is undoubtedly high. The low specificity of the criteria weakens the evidence.

Lines 30 to 31: "These findings highlight the complex etiology of stunting and suggest nutritional, and other biomedical interventions are insufficient" Wouldn't the basic causes of malnutrition be the same as for SRQ +? Thus, preventive measures should not be based on the basic causes in common?

Lines 31 to 32: "… promotion of mental and behavioral health programs for parents are essential to achieve child growth and development". What are the appropriate measures for promoting mental health?

Lines 43 to 45: "a recent study added fetal growth restriction and preterm birth as risk factors and suggested they account for as much as 32% of stunted children". Obviously, the contribution of different risk factors to height deficit varies significantly according to the peculiar characteristics of the respective scenarios where the study has been carried out. Besides, the prevalence of LBW itself (low or high prevalence) will significantly influence this participation.

Lines 46 to 48: "However, these risk factors combined still cannot account for even a majority of stunted children. The etiology of restricted linear growth remains poorly understood (3, 8, 9)." This statement needs a little more caution. Regardless of the underlying causes, growth retardation ultimately results from deprivation of nutrients and energy at the cellular level for prolonged periods, which is why stunting in children is assumed to be an indicator of chronic undernutrition at the population level. And this results from the interaction between several factors.

Line 49: "Because most studies have been limited to biomedical causes of growth restriction," Literature is abundant in publications that address non-biomedical causes.

Lines 50 to 53: "Factors such as parental psychological distress could diminish the quality of caregiving behaviors and enhance psychological stress for the child, both of which may affect growth via the hypothalamic-pituitary-adrenal (HPA) axis and other pathways." It would be interesting to explore these aspects further in the discussion.

Lines 55 to 58: "Psychological distress assessment in India and Vietnam using the Self Reporting Questionnaire (SRQ20), found that high maternal mental disorder contributed to child stunting and underweight status. Another study in Bangladesh and Vietnam found maternal mental disorder was related to child stunting and underweight". Probably because they are derived from the same basic cause: low socioeconomic conditions.

Lines 63 to 64: "As such, their findings may have limited relevance to the general population and are subject to confounding." In this regard, care should be taken with very large samples, as they show statistical significance when the finding has no clinical significance. For example, how relevant is a deficit of 0.091 (95% CI: -0.18; -0.0033) z-score?

Lines 86 to 87: "For our study, we selected data from all households comprised of two-parent families with children age 6 to 59 months, and with data on parental distress". The lack of a partner has been reported as an important risk factor for both mental health and stunting. This exclusion probably caused an underestimation in both outcomes.

Lines 95 to 96: "HAZ scores were classified as mild, moderate, and severe stunting using cut off points of < -1 HAZ, <-2 HAZ, and <-3 HAZ, respectively". It is not reasonable to attribute stunting to children between -1 and -2 z scores.

Lines 107 to 108: "We also built a binary distress variable using 'no parental distress' and 'any parental distress' at least distress score = 1". Distress score = 1: does this mean that among the 20 SRQ questions, the individual answered at least one of them positively? If so, is there any evidence in the SRQ validation studies regarding the use of this form? Wouldn't the specificity of the indicator become too low?

Lines 112 to 114: "Physiological factors were mid-upper arm circumference (MUAC), body mass index (BMI), height and age of the mother, height and age of the father, and child sex". These variables are not physiological but anthropometric and demographic.

Lines 114 to 115: "Health behavior factors consisted of used of iodized salt, drinking water source, garbage disposal, water waste disposal, stool disposal, …" Some of these variables would be better classified as access to service infrastructure.

Lines 123 to 125: "The multivariate-adjusted analyses incorporating sampling weights were conducted using linear, logistic, and multilevel multinomial logistic regression with PSU as a random effect." The dependent and independent variables are autocorrelated. How was the possibility of multicollinearity addressed? Wouldn't principal component analysis/factor analysis make the study more robust?

Lines 125 to 126: "We determined the full and the best fitting model for the relative risk of stunting associated with parental distress and other covariates". Relative risk is an incidence ratio. If the study was cross-sectional, the frequency measure should be a prevalence ratio or odds ratios (but, in this study, OR would not be recommended considering that the dependent variable was greater than 10% and thus would overestimate the association).

Lines 145 to 146: "The proportion of mothers, fathers, and parents with any distress (i.e. distress of at least 1) were 39.6%, 33.3%, and 51.6%, respectively." Has the SRQ been validated for use in this way (classify as positive individuals with only one answer yes?) This gives a false impression of a high magnitude of the problem.

Line 147: “classified as distressed (i.e. SRQ score of at least 6) were 4.1%, 2.6%, and 0.9%, respectively.” The difference between the prevalences obtained with the two approaches is very different. The first strategy creates in the reader a distorted image as to the real magnitude of the problem. Better to use the SRQ as it was validated.

Table 1 To "clean up" the table and make it more intelligible, I suggest removing the N from the categories, informing only the total N of the variable and, for the categories, only the respective percentages.

Lines 180 to 181: "Physiological factors related to stunting were child sex (boy), low maternal MUAC, older maternal and paternal age, paternal and maternal short stature." As already mentioned, these variables are not physiological.

Lines 260 to 262: "Based on the population of Indonesia in 2013, this would account for approximately 14.65 million stunted children. And 51.6% of under-five children experienced a parent with any level of distress- either maternal (39.6%) or paternal (33.3%) or both (21.3%);" These values are overestimated. In the first case, due to the inclusion of a high proportion of individuals distributed following the expected frequency in a healthy population and, in the case of distress, by the use of criteria in disagreement with what was proposed in the original method.

Lines 276 to 277: "Paternal and parental distress were associated with reductions of HAZ by 0.13, and 0.23 z-scores, respectively." Usually, mothers are the main responsible for the care of the child. In the analyzed sample, this should be made even more evident by the fact that the vast majority of fathers, unlike mothers, have work outside the home. So how explain that paternal distress contributes to growth deficit, but maternal distress does not?

Lines 320 to 322: "We recommend promotion of mental and behavioral health for parents, which would promote child growth and also promote early childhood cognitive development". What would be the causes of parental distress? So, what actions should be taken to address this problem?

Lines 323 to 326: "Perhaps most striking is the observation that parental distress interacting with multiple other factors accounted for 5.6% of the overall loss of HAZ-score, virtually all of its effect, rendering it amongst the larger factors in poor linear growth in this analysis. This highlights the large contribution of psychosocial factors to linear growth restriction via other factors". This argument is difficult to understand: is not 5.6% the part of the loss in HAZ explained by the parental distress? If so, the last sentence does not exaggerate the importance given to distress, given that 94.4% of the growth restriction would be explained by other factors? Please clarify.

Lines 336 to 340: "Behavior modification interventions should improve child growth by targeting improved waste water disposal, solid waste disposal, septic tank use, handwashing, and quitting smoking. While intervention on social conditions include increasing household income, improving maternal education, providing jobs, and attention to rural mothers' conditions and working mothers." Wouldn't these actions, in addition to preventing growth deficit, also be effective in reducing the prevalence of distress?

Lines 353 to 358: "Concerns regarding maternal distress in relation to child growth have been rising, and a better understanding is needed of interventions for maternal, paternal and parental distress. This would include more studies to understand how maternal, paternal and parental distress could reduce linear growth of the children, and therefore exacerbate the poor growth of the children. If this finding is confirmed, promotion of mental and behavioral health programs must to be pursued as part of a comprehensive strategy to enhance child growth and development." In addition to the harm in caring for the child, would any pathophysiological mechanisms be interfering with the growth process? In the introduction, the participation of the hypothalamic-pituitary-adrenal axis and other pathways, and changes in nutrient metabolism or immune function, was mentioned. Wouldn't it be worth discussing a little more about these mechanisms?

Lines 364 to 368: "As such, while the cumulative loss of height contributed from parental distress itself was limited, the effects of parental distress on overall linear growth via indirect effects is substantial. Given that multiple behavioral and socioeconomic factors influence stunting, we suggest promotion of mental and behavioral health programs for parents to promote child growth and child development." Then, wouldn't it be more suitable to pay attention to the causes here called of indirect effects? Based on the reported findings, it would not be more appropriate to propose broader and more integrated actions on all determinants or, at least, relativize more prudently the role of distress? Caution is needed regarding the external validity of the study. The outcomes may vary according to the characteristics of the scenarios where the studies are developed. By the way, let's see the conclusion presented in the study by Harpham et al. [6]: "There was a relation between high maternal CMD and poor child nutritional status in India and Vietnam. However, the findings from Peru and Ethiopia do not provide clear evidence for a similar association being present in non-Asian countries. Regardless of the direction of the relation, child nutrition programmes in Asia should consider incorporating promotion of maternal mental health."

1. Carvalhaes MA, Benicio MH: [Mother's ability of childcare and children malnutrition]. Revista de saude publica 2002, 36(2):188-197.

2. Girma S, Fikadu T, Abdisa E: Maternal Common Mental Disorder as Predictors of Stunting among Children Aged 6-59 Months in Western Ethiopia: A Case-Control Study. International journal of pediatrics 2019, 2019:4716482.

3. Hassan BK, Werneck GL, Hasselmann MH: Maternal mental health and nutritional status of six-month-old infants. Revista de saude publica 2016, 50:7.

4. Nguyen PH, Saha KK, Ali D, Menon P, Manohar S, Mai LT, Rawat R, Ruel MT: Maternal mental health is associated with child undernutrition and illness in Bangladesh, Vietnam and Ethiopia. Public health nutrition 2014, 17(6):1318-1327.

5. Santos DS, Santos DN, Silva Rde C, Hasselmann MH, Barreto ML: Maternal common mental disorders and malnutrition in children: a case-control study. Social psychiatry and psychiatric epidemiology 2011, 46(7):543-548.

6. Harpham T, Huttly S, De Silva MJ, Abramsky T: Maternal mental health and child nutritional status in four developing countries. Journal of epidemiology and community health 2005, 59(12):1060-1064.

7. Meza JM: [Prevalence of mental health problems in parents of children with malnutrition in one area of Tegucigalpa, Honduras]. Acta psiquiatrica y psicologica de America latina 1988, 34(2):145-148.

8. United Nations Children's Fund: Strategy for improved nutrition of children and women in developing countries. The Indian Journal of Pediatrics 1991, 58(1):13-24.

6. PLOS authors have the option to publish the peer review history of their article (what does this mean?). If published, this will include your full peer review and any attached files.

Reviewer #1: Yes: Isabel Giacomini Marques

Reviewer #2: Yes: HAROLDO DA SILVA FERREIRA

---

## [Author Response · Author response to Decision Letter 0]

8 Jan 2021

Dear Editor and Reviewers,

We would like to thank you for arranging peer-review of our manuscript and for your invitation to submit a revised version. We appreciate the effort of the reviewers, and believe that their constructive suggestions have resulted in a stronger manuscript for PLOS ONE’s readers. We have made substantial changes (indicated in track changes in the manuscript) to address the Editor and Reviewers

Here we addressed comment of editor and reviewers:

Journal Requirements:

Response: 

We tried to follow PLOS ONE's style requirements

1) Please provide additional details regarding participant consent. In the ethics statement in the Methods and online submission information, please ensure that you have specified whether consent was informed.

Response: 

We have added information about participation consent in the methods section and online submission.

2) Please improving statistical reporting and refer to p-values as "p<.001" instead of "p=.000". Our statistical reporting guidelines are available at https://journals.plos.org/plosone/s/submission-guidelines#loc-statistical-reporting

Response: 

We have changed the p-values as “p<0.001” instead of “p=0.000”.

3) Please do not include funding sources in the Acknowledgments or anywhere else in the manuscript file. Funding information should only be entered in the financial disclosure section of the submission system. https://journals.plos.org/plosone/s/submission-guidelines#loc-acknowledgments. 

Response: 

We have removed funding sources from the acknowledgments.

4) We note that you have stated that you will provide repository information for your data at acceptance. Should your manuscript be accepted for publication, we will hold it until you provide the relevant accession numbers or DOIs necessary to access your data. If you wish to make changes to your Data Availability statement, please describe these changes in your cover letter and we will update your Data Availability statement to reflect the information you provide.

Response: 

We have revised our cover letter indicates that data will not be available. We obtained the data from National Institute of Health Research and Development, Ministry of Health Republic of Indonesia. We consider the institution permits data utilization for this paper only, for further exploration of the data need legal consent from the institution. Thus, we will not upload our raw data in the website.

5) Please include captions for your Supporting Information files at the end of your manuscript, and update any in-text citations to match accordingly. Please see our Supporting Information guidelines for more information: http://journals.plos.org/plosone/s/supporting-information

Reviewers' comments:

Reviewer's Responses to Questions

Comments to the Author

1. Is the manuscript technically sound, and do the data support the conclusions?

Reviewer #1: Yes

Reviewer #2: Partly

2. Has the statistical analysis been performed appropriately and rigorously?

Reviewer #1: Yes

Reviewer #2: Yes

3. Have the authors made all data underlying the findings in their manuscript fully available?

Reviewer #1: Yes

Reviewer #2: No

4. Is the manuscript presented in an intelligible fashion and written in standard English?

Reviewer #1: No

Reviewer #2: Yes

5. Review Comments to the Author

Reviewer #1: The present study has important findings and highlights the need of a comprehensive attention to the child development. The findings of this study provide a good description of the studied population and present a range of “not obvious” factors that may have an essential role to an adequate child development. Additionally, the discussion promotes a good dialogue between the findings of this study and the literature.

Response: 

We would like to thank Reviewer #1 for believing our study has important findings and highlights the need of a comprehensive attention to the child development, and also other positive evaluations. We appreciate the positive evaluations, that could support us to revise this manuscript. 

The introduction brings a great set of information about the topic and provides a contextualization of the subject to the reader, but also signalizes a gap in the literature about the theme, highlighting the potentialities and limitations of previous studies. However, in this section, there are some information without scientific references. For example, in the beginning of the manuscript, long- and short-term effects of the linear growth restriction (line 36-39) were point out, but there are no references. Next, the text quotes “multiple studies (…)” (line 41-43) and a “recent study” (line 43), but don’t provide the references of these studies.

Response: 

Thank you. We have added the references. We add references [1-4] for line 36-40, reference [4-5] for line 40-42 and reference [6, 10-15] for line 46-49.

Still in the introduction, there are some statements that are confusing, may leading to misinterpretation. For example, in the phrase “Additional risk in some studies include environmental factors, maternal health status, and child health status” (line 45), it is not clear what is in risk and how these factors where explored in each article.

Response: 

In order to minimize misinterpretation, we revised the sentence into “Additional risk of stunting in some studies include environmental factors, maternal health status, and child health status”. The term risk of stunting states in the next sentence. Each factor has been explored in the covariates we used in the analysis.

The methodology section evidences the power of this study, proportionating good perspective of the study design to the reader and evidencing the large data set used. Nevertheless, some aspects should be reviewed. In the variables section, you stablish the cut off points to classify stunting as mild, moderate and severe (line 95-96), but it would be important to reference it (i.e. WHO child growth standards). Also, the creation process of the distress variables, using different score cut-off points, is not clear in the text (line 101-108). It was only possibly to fully understand the categorization of the variables when observing the tables.

Response: 

Thank you for your suggestion, we have added reference from (Stevens et al., 2012) and (Agho et al., 2019) in line 108. We include mild stunting to give initial alarming on restricted linear growth. In addition, the hazardous effect of stunting could happen along in a continuum of mild, moderate, and severe malnutrition. The PAR (population attributable risk) estimated 83% child malnutrition related deaths were attributable to mild to moderate, rather than severe malnutrition (Pelletier et al., 1995). The later analysis (Black et al., 2008) show mild malnutrition contribute to diseases mortality among under five children. Although mild stunting did not show significant. However, in our analysis showing significant contribution of maternal, paternal and parental distress on mild and moderate stunting, but was not showing significant association to severe stunting. 

Response: 

We also added explanation the creation process of the distress variables as follows: 

We calculated maternal and paternal distress scores by summing the item scores, which ranged from 0 to 20 (0 indicating ‘no distress' to 20 indicating 'severe distress'). We categorized maternal and paternal distress into two binary variables using a cut-off point of 6 [18] and 1. A cut off point of <6 indicated ‘no distress’, and ≥6 indicated ‘distress’. From this binary variable, we created four parental distress categories: both the mother and father with no distress, only the mother with distress, only the father with distress, and both the mother and father with distress. 

We deleted variable ‘any parental distress’ as also reviewer#2 questioned, because of no reference supported the cut-off point. And we did re-analyses to address several reviewer comments (line 109-119).

Another aspect to be reviewed is the nature of the variables. You describe MUAC, BMI, height and age of the mother/father and child age as physiological factors (line 112-114), but these variables should be placed in distinct groups since their nature is different. It is important to note that physiological conditions are related to organic and biologic aspects of the human body. Next, you describe iodized salt, drinking water source, garbage disposal, water waste disposal, stool disposal as health behaviours (line 114-115). However, behaviours are more connected to individual actions then sanitation conditions and health access.

Response: 

The MUAC, BMI, and height was anthropometric measurement. Anthropometric measurements are a series of quantitative measurements of the muscle, bone, and adipose tissue used to assess the composition of the body. Thus, we categorized those variables as physiological factors.

We highlight the variables in the study into four groups including disease, physiological, health behavior and socio-economic factors for both child and parents. While a review of child stunting determinants in Indonesia did not clearly highlight distinct between child and parent physiological factors (Beal et al., 2018). We did not distinct those factors for several reasons eg. (1) the biological aspect of child and parent has similar genetics, neural structure and functions, and hormones. (2) distinct those factors did not add significant values in determining factors on stunting.

We agreed that health behaviors are more connected in to individual actions; However, we included those variables in the behavior group with reasons i.e. (1) in the questionnaires ask about the use of sanitation facilities, (2) iodized salt, improved drinking water source, garbage disposal, water waste disposal, and stool disposal were the facilities that belong to families, (3) there are individual behavior in the family that to use or practice for those belonging, including parents.

Still in the methodology, you describe the formula used to obtain the proportion of z-score lost due to any parental stress. However, the text is a bit confusing and not clear, making difficult to understand the construction this formula (line 128-132).

Response: 

We have added more formulas and provide explanation in order to give better understanding on the proportion of z-score lost due to any parental distress. 

We present the beta coefficient of HAZ score and proportion of z-score lost due to any parental distress and other risk factors. For the proportion of z-score lost due to any parental stress, the numerator was the sum of the products for each child of the risk beta coefficient times the Y value, and the denominator was the sum of the z-score lost for each child based on the intercept and sum of the product of significant beta values of each covariate and the value of the response variable.

% Z score lost of risk factor Y=(sum of the product for each child of Y risk beta HAZ x Y score)/(sum of intercept of each child+ sum of each child of all risks (beta HAZ x Y score)) x 100

The results from the study are described in a very fine way. However, there are some important considerations in order to contribute to the quality of this section. In this segment you affirm that maternal, paternal and parental distress are associated with lower HAZ according to the figure 3 (line 153-154). Even though this figure brings information in a really clear way, there is no data about the p value in this figure.

Response: 

Thank you for your suggestion, the p values have been added in the Figure 3. Figure 3 illustrates line plots using HAZ of maternal, paternal, and parental distress score instead of regression line for Table 2. We used p value of Table 2 for Figure 3. 

Next, you say that parental distress accounted for a reduction of 0.36 HAZ (line 156-157), but this information is not available in the table 2.

Response: 

Thank you for your correction, this has been revised into parental distress accounted for a reduction of 0.19 HAZ (line 168-170).

Still in the results section, you say that paternal hight was associated with mild, moderate and severe, but don’t mention clearly if you are referring to stunting (line 188).

Response: 

Thank you for your correction, we revised the sentence into “Paternal height less than 155cm was associated with mild (RR=1.22, 95%CI 1.10 to 1.35; p<0.001), moderate (RR=1.62, 95%CI 1.47 to 1.80; p<0.001) and severe (RR=1.81, 95%CI 1.63 to 2.00; p<0.001) stunting.” Line 197-202.

The discussion of the findings highlights important previous findings, intertwining the existing literature and the findings from the presented study. Yet, considerations must be taken in account to improve this section. In the beginning of the discussion you state that parental distress was associated with linear growth of their children. However, this is a cross sectional study, making it not possible to track linear growth. Also, there is an extensive description of the results in the first paragraph, not fitting into the discussion section requirements. On the other hand, some studies mentioned are not well described, making the dialogue between previous findings and the results not comparable (i.e. references number 32 and 34, 36, 37, 4, 40-44 not indicating the age of the studied population or methods). There is also information without references (line 302-305 and line 349-352).

Response: 

We aware that this is a cross sectional study, in this study we highlight linear growth of the children based on ages. Thus, we add this issue in the limitation section. However, cross-sectional data still can be used to measure linear growth. As the WHO multicenter growth reference study group (2006) using both longitudinal and cross sectional data and highlight appropriateness to be use for constructing international growth standard (WHO Multicentre Growth Reference Study Group, 2006).

We thank for your comments that description of the result in the first paragraph did not fit into the discussion section. In this paragraph we intend to discuss about the importance of parental distress and stunting in Indonesia context. We also would like to raise awareness substantial burden of stunting and parental distress in Indonesia. Thus, we separated into two paragraphs as follows (line 299-314):

“The present study found that parental distress was associated with reduced linear growth of their children. More than half of under-five children were mildly, moderately, or severely stunted. Based on the population projection of Indonesia in 2013 there were approximately 23,994,200 under-five children (Statistics Indonesia, 2013), this would account for 15.31 million stunted children. And 7.5% of under-five children experienced parental distress - either maternal (4.1%) or paternal (2.6%) or both (0.9%). Of those children exposed to parental distress, approximately two-thirds (70.4%) had mild, moderate, or severe stunting. Based on these data we estimate approximately 1.27 million stunted children in Indonesia reside in households experiencing parental distress. This is a substantial burden that is likely to be underestimated. We note the proportion of households with psychological distress from the Self Reporting Questionnaire (SRQ20) at the provincial level ranged from 1.6% to 15.9%, with urban centers tending toward higher stress levels, suggesting this burden is likely to increase as urbanization is increasing globally. Under-five children are in a critical stage of development, needing stimulation and attention (Prado et al., 2017, McDonald et al., 2016). Distress among parents may affect parenting quality, especially responsiveness to their interactions, or exacerbate toxic stress that is known to affect child growth and development (DiPietro et al., 2006, Kiernan and Huerta, 2008).”

Thank you. We have revised the paragraph as follows (264-275):

“In this study, we found that for each one-unit increase in the score for paternal and parental distress, the HAZ of the child decreased by 0.11 and 0.19, respectively. This reduction is higher than the HAZ reduction related to maternal distress alone. Paternal distress has been commonly known to be related to low income and poverty, job insecurity, and social inequalities (World Health Organization, 2012). We were unable to find a previous study that yielded the relationship between parental and paternal distress on stunting or growth of children. Our study also indicates that maternal distress was associated with the reduction of HAZ by 0.09, and this is larger than previously reported by other studies in rural Bangladesh where infants at age 12 months whose mothers had symptoms of depression had a 0.007 drop in mean HAZ (Black et al., 2009). A longitudinal study of a nationally representative US birth cohort showed mothers with mild and moderate depressive symptoms had children who were, on average, 0.26 cm shorter in stature (95% CI: -0.48 cm to -0.05 cm) than their peers (Surkan et al., 2014).”

The limitations and potentialities of the study are mentioned in the manuscript. However, it is important to highlight that this study did not include the one-parent families, which may be a considerable part of the initial households and may present different results.

Response: 

Thank you for your suggestion, we have added this limitation into the manuscript (line 365-368): “However, this study has limitations. First, this was a cross sectional study which could not infer causality. Second, this study did not include the one-parent families, which may be a considerable part of the initial households and may have negative mental health consequences (Benzeval, 1998).”

In addition, since we analyzed the paternal risk factors so if the observations didn’t have paternal data, these will be excluded. 

An important caveat to point about the text concerns to the morbidity variable. When first describing the morbidity variable in the methodology, you say that “upper respiratory infection” was considered a disease (line 111), but latter you replace it with “acute respiratory infection” (line 177), which is not a synonym, but a different condition. In addition, in the results section you affirm that infectious disease was related to stunting, but then you include a condition that can be originated by an infection or not (ie. diarrhea) in this variable (line 177-178). Also, in table 2 there is only the variable “disease”, not presenting different results to each one of the nature of the disease (non-infectious versus infectious disease).

Response: 

The” upper respiratory infection” in line 111 should be “acute respiratory infection”, thus we have replaced it and make it consistent through the paper. We also added infectious disease in the table 2.

The revised version of line 119-122 as follows:

“Other covariates of interest included infectious disease, physiological, health behavior, and socioeconomic factors of the child and parents. Infectious disease factors consisted of binary variables indicating whether or not the child suffered from diarrhea, acute respiratory infection, pneumonia, or malaria in the last month.”

Overall, the manuscript is well structured and organized. Yet, the language quality should be verified, since many parts demands extra attention to interpret the information. Also, the language may compromise the understanding of the content, leading to ambiguity and doubts.

Response: 

We had check revised some sentence that may lead to ambiguity. 

Reviewer #2: The psychological distress of parents is associated with reduced linear growth of children: evidence from a nationwide population survey

Review PONE-D-20-08335

The article aims "to identify whether the psychological distress of parents is related to child linear growth and stunting, and to document the associated risk factors, and examine the relationship between parental distress and other behavioral risk factors for stunting", for which analyzed secondary data (n=54,261 households) from the Indonesia National Health Survey 2013.

This is an interesting topic for public health, although the demonstration of the relationship between mental health and nutritional status of children is not new [1-7]. As strengths of the study could be cited: (a) the use of a nationally representative sample involving more than 50,000 families, and (b) the indication that parental distress, interacting with multiple other factors, accounted for 5.6% of the overall loss of HAZ-score. The main negative points were: (a) use of outcome classification criteria favoring sensitivity, at the expense of specificity, which generates many cases of false positives, weakening the strength of the evidence by overestimating the findings; (b) Overvaluing the results found in the unadjusted analysis, when it should focus on data from the adjusted analysis, when, in theory, the effects of confounding factors were reduced. In this aspect, numerous studies indicate that mental health disorders are associated with several risk factors common to undernutrition. Thus, the importance of this study would be to separate the specific contribution of this outcome in determining child growth deficit; (c) In order to value the theme explored (psychological distress of parents in the etiology of stunting), the authors minimize the role of socio-economic-demographic aspects. It is not true that "Although the etiology is recognized to be complex, most analyses have focused on biomedical determinants, with limited attention to social factors affecting care and nurturing in the home". In the old and well-known model for determining undernutrition, published by UNICEF [8], structural causes, such as political, economic and ideological structure, are recognized as basal (distal) determinants, preceding underlying causes and immediate causes. There is consensus that any interventions to address the problem of undernutrition in a population will not be sustainable if constant and permanent access to education, health care, adequate sanitation conditions and healthy food is not promoted.

Response: 

We would like to thank Reviewer #2 for the comments and thoughtful critiques. We have incorporated your specific comments into a revised version of the manuscript, and below we have addressed your comments each one individually.

More specifically, we present the following comments:

Lines 4 to 6: "Although the etiology is recognized to be complex, most analyses have focused on biomedical determinants, with limited attention to social factors affecting care and nurturing in the home". See comment above.

Response: 

It is true that the importance of social factors affecting care and nurturing in the home have been addressed by UNICEF as well as several publications (Fund, 1991, Beal et al., 2018). We revised the sentence as follows (line 4-6)

“Although the etiology is recognized complex, most analysis have focused on social and biomedical determinants, with limited attention on psychological factors affecting care and nurturing in the home”. 

Lines 18: "paternal distress increased the risk of mild stunting (HAZ <-1) by 38% (95% CI 1.21,1.57)"… The proportion of children below - 1 sd in the healthy population (anthropometric pattern) is over 33%. Is it prudent to consider these children as having stunting? The rate of false positives is undoubtedly high. The low specificity of the criteria weakens the evidence.

Response:

The World Health Organization has been categorized HAZ<-2 as stunting(WHO Multicentre Growth Reference Study Group, 2006, World Health Organization, 2019). However, some studies highlighted the hazardous effect of malnutrition could happen along in continuum of mild, moderate, and severe category (Stevens et al., 2012, Agho et al., 2019). We aware of high false positive if we categorize those three categories into a single category of stunting. Thus, in our analysis were divided into three categories and the mild stunting was created to give initial alarming on restricted linear growth. We also use the continuous variable of HAZ to minimize bias of categorization and the result can be seen in table 2 (Das-Smaal, 1990).

We aware that the proportion of children bellow -1 SD in healthy population was high, this could give notification of high burden of children that can easily drop down into moderate and severe malnutrition. Furthermore, a study revealed the PAR (population attributable risk) estimated 83% child malnutrition related deaths were attributable to mild to moderate, rather than severe malnutrition (Pelletier et al., 1995). The later analysis show mild malnutrition contribute to diseases mortality among under five children (Black et al., 2008). Although mild stunting did not show significant. However, in our analysis showing significant contribution of maternal, paternal and parental distress on mild and moderate stunting, but only parental distress was showing significant association to severe stunting.

To address your comments, we added these explanations in the methods sections line 106-108.

“We included mild stunting (< -1 HAZ) to give initial alarming on restricted linear growth. Since, the hazardous effect of stunting could happen along in a continuum of mild, moderate, and severe malnutrition [33]”.

Lines 30 to 31: "These findings highlight the complex etiology of stunting and suggest nutritional, and other biomedical interventions are insufficient" Wouldn't the basic causes of malnutrition be the same as for SRQ +? Thus, preventive measures should not be based on the basic causes in common?

Response:

Yes, it is true. Basic causes of both malnutrition as well as mental health could be the same such as socioeconomic and environmental factors (Stewart, 2007, Beal et al., 2018). In order to reduce stunting, nutritional and other biomedical intervention might be insufficient. Thus, we suggest different preventive factors to be considered ie. parental mental health and health behavior. However, it does not mean to neglect those previous factors. 

Lines 31 to 32: "… promotion of mental and behavioral health programs for parents are essential to achieve child growth and development". What are the appropriate measures for promoting mental health?

Response:

The Ottawa Charter of Health Promotion highlight five action strategies for health promotion, including build healthy public policy, create supportive environment, strengthen community action, develop personal skill and reorient health services (Organization, 1986). A review explored actions for each strategy(Jané-Llopis et al., 2005). In addition, a study revealed that mental health promotion should be works at three levels: strengthening individuals, strengthening communities and reducing structural barriers to mental health. Thus, we add the statement as follows (30-33):

“......programs for parents must to be pursued as part of a comprehensive strategy to enhance child growth and development, i.e. improve stakeholder capacity, integrated community development, improve parenting skills, as well as reduce gender discrimination, and domestic violence”

Lines 43 to 45: "a recent study added fetal growth restriction and preterm birth as risk factors and suggested they account for as much as 32% of stunted children". Obviously, the contribution of different risk factors to height deficit varies significantly according to the peculiar characteristics of the respective scenarios where the study has been carried out. Besides, the prevalence of LBW itself (low or high prevalence) will significantly influence this participation.

Response:

It's true that the particular characteristics of the study significantly influence the contribution of different risk factors to height deficit. Thus, we use a study that pooled data from 137 developing countries, including Indonesia (Danaei et al., 2016). Please see result of that study on growth’ at table 2 in first row under column heading ‘Fetal growth restriction and fetal showing 32%.

Lines 46 to 48: "However, these risk factors combined still cannot account for even a majority of stunted children. The etiology of restricted linear growth remains poorly understood (3, 8, 9)." This statement needs a little more caution. Regardless of the underlying causes, growth retardation ultimately results from deprivation of nutrients and energy at the cellular level for prolonged periods, which is why stunting in children is assumed to be an indicator of chronic undernutrition at the population level. And this results from the interaction between several factors.

Response:

We revised the sentence as follows (line 47-49):

“However, these risk factors combined still insufficient to account of stunted children. Thus, the etiology of restricted linear growth needs deep exploration for better understanding (Danaei et al., 2016, Rakotomanana et al., 2016, Bhutta et al., 2017)..”

Line 49: "Because most studies have been limited to biomedical causes of growth restriction," Literature is abundant in publications that address non-biomedical causes.

Response:

We revised the sentence as follows (line 50-51): “Limited study explored the impact of household socio-emotional risk factors of growth restriction.”

Lines 50 to 53: "Factors such as parental psychological distress could diminish the quality of caregiving behaviors and enhance psychological stress for the child, both of which may affect growth via the hypothalamic-pituitary-adrenal (HPA) axis and other pathways." It would be interesting to explore these aspects further in the discussion.

Response:

We agreed, we have added in the discussion section (line 276-281).

“Our analyses indicate that parental distress is related to mild and moderate stunting, but not severe stunting. Previous study suggested that maternal stress and/or depression may affect disturbance of the hypothalamic-pituitary-adrenal (HPA) axis thus exhort physiological effect during intrauterine(Field et al., 2006). This mechanism effect on fetal weight which mediated by prenatal cortisol and norepinephrine levels. Maternal distress also affects inadequate nutritional care.”

Lines 55 to 58: "Psychological distress assessment in India and Vietnam using the Self Reporting Questionnaire (SRQ20), found that high maternal mental disorder contributed to child stunting and underweight status. Another study in Bangladesh and Vietnam found maternal mental disorder was related to child stunting and underweight". Probably because they are derived from the same basic cause: low socioeconomic conditions.

Response:

Thank you. The paragraph was edited (line 62-66). "Psychological distress assessment in poor and middle incomes areas in India and Vietnam using the Self Reporting Questionnaire (SRQ20), found that high maternal mental disorder contributed to child stunting and underweight status. Another study in Bangladesh and Vietnam where poverty; malnutrition; and poor mental health coexist, found maternal mental disorder was related to child stunting and underweight (Nguyen et al., 2018)."

Lines 63 to 64: "As such, their findings may have limited relevance to the general population and are subject to confounding." In this regard, care should be taken with very large samples, as they show statistical significance when the finding has no clinical significance. For example, how relevant is a deficit of 0.091 (95% CI: -0.18; -0.0033) z-score?

Response:

We aware that analysis in the very large samples tend to be statistically significant. In this analysis however we can see the gradation of the effect from maternal, paternal and parental distress. In addition, reduction of 0.09 z-score may have no clinical significance, but if we compare to other factors such as maternal education or paternal smoking behavior, the effect of maternal, paternal and parental distress is considerable.

This was larger than previously reported by a study in rural Bangladesh where infants at age 12 months whose mothers has symptoms of depression had a 0.007 drop in mean HAZ (Black et al., 2009).

Lines 86 to 87: "For our study, we selected data from all households comprised of two-parent families with children age 6 to 59 months, and with data on parental distress". The lack of a partner has been reported as an important risk factor for both mental health and stunting. This exclusion probably caused an underestimation in both outcomes.

Response:

We have added in the limitation of this study in line 364-366. 

“Second, this study did not include the one-parent families, which may be a considerable part of the initial households and may have negative mental health consequences (Benzeval, 1998).”

In addition, since we analyzed the paternal risk factors so if the observations didn’t have paternal data, these will be excluded. We compared, there were no significant difference between single-parent and two-parent families on stunting prevalence.

Lines 95 to 96: "HAZ scores were classified as mild, moderate, and severe stunting using cut off points of < -1 HAZ, <-2 HAZ, and <-3 HAZ, respectively". It is not reasonable to attribute stunting to children between -1 and -2 z scores.

Response:

As we have address in the previous comment, mild stunting classification was created to give initial alarming on restricted linear growth. And we added this explanation in the methods section line 106-108 as follows:

“We included mild stunting (< -1 HAZ) to give initial alarming on restricted linear growth. Since, the hazardous effect of stunting could happen along in a continuum of mild, moderate, and severe malnutrition(Stevens et al., 2012)”

Lines 107 to 108: "We also built a binary distress variable using 'no parental distress' and 'any parental distress' at least distress score = 1". Distress score = 1: does this mean that among the 20 SRQ questions, the individual answered at least one of them positively? If so, is there any evidence in the SRQ validation studies regarding the use of this form? Wouldn't the specificity of the indicator become too low?

Response:

Thank you, we deleted this variable from our analysis.

Lines 112 to 114: "Physiological factors were mid-upper arm circumference (MUAC), body mass index (BMI), height and age of the mother, height and age of the father, and child sex". These variables are not physiological but anthropometric and demographic.

Response:

As we have addressed to the reviewer #1, we stated those variables as physiological factors by merging the physiological factors for both child and parents. It is true that MUAC, BMI, and height was anthropometric measurement. Anthropometric measurements are a series of quantitative measurements of the muscle, bone, and adipose tissue used to assess the composition of the body (Casadei and Kiel, 2020). Thus, we categorized those variables as physiological factors. There are abundant articles using this term for those variables.

Lines 114 to 115: "Health behavior factors consisted of used of iodized salt, drinking water source, garbage disposal, water waste disposal, stool disposal, …" Some of these variables would be better classified as access to service infrastructure.

Response:

These variables were not access to service infrastructure, but mainly the behavior household on dispose stool, garbage or water waste. 

Lines 123 to 125: "The multivariate-adjusted analyses incorporating sampling weights were conducted using linear, logistic, and multilevel multinomial logistic regression with PSU as a random effect." The dependent and independent variables are autocorrelated. How was the possibility of multicollinearity addressed? Wouldn't principal component analysis/factor analysis make the study more robust?

Response:

We have tested the multicollinearity by estimating the variance inflation factors (VIF) and using a tolerance of a VIF value less than 10 (Hair et al., 2014). But in often the case in logistic regression, values above 2.5 may be a cause for concern (Shadfar and Malekmohammadi, 2013). From our analysis, we found that the mean VIF was 1.14 and each variable has VIF less than 2. This best fit model considered to be moderate correlation. Thus, this analysis is robust to estimates the coefficient. 

Collinearity Diagnostics 

Variables VIF

Nutritional status 1.04

Residence 1.23

Paternal occupation 1.18

Maternal occupation 1.06

Maternal education 1.36

Number of house member 1.08

Wealth quintile 1.98

Paternal smoking 1.03

Improved water waste disposal 1.06

Garbage disposal 1.11

Used of latrine 1.47

Paternal age 1.11

Paternal height 1.03

MUAC 1.04

Maternal BMI 1.01

Maternal height 1.04

Sex of the child 1.00

Iodized salt used 1.02

Morbidity 1.01

Parental distress score 1.01

Mean VIF 1.14

Lines 125 to 126: "We determined the full and the best fitting model for the relative risk of stunting associated with parental distress and other covariates". Relative risk is an incidence ratio. If the study was cross-sectional, the frequency measure should be a prevalence ratio or odds ratios (but, in this study, OR would not be recommended considering that the dependent variable was greater than 10% and thus would overestimate the association).

Response:

Thank you. Relative risk ratios (RRR) can be interpreted in a similar manner to odds ratios in the ordinary logit model. They are merely the exponentiated multinomial logistic model (MLM) coefficients from the regression output. Some cross-sectional studies use RRR (Lv et al., 2011, Bham et al., 2012, Camey et al., 2014)

Lines 145 to 146: "The proportion of mothers, fathers, and parents with any distress (i.e. distress of at least 1) were 39.6%, 33.3%, and 51.6%, respectively." Has the SRQ been validated for use in this way (classify as positive individuals with only one answer yes?) This gives a false impression of a high magnitude of the problem.

Response:

We have removed from our analyses.

Line 147: “classified as distressed (i.e. SRQ score of at least 6) were 4.1%, 2.6%, and 0.9%, respectively.” The difference between the prevalences obtained with the two approaches is very different. The first strategy creates in the reader a distorted image as to the real magnitude of the problem. Better to use the SRQ as it was validated.

Response:

Thank you. We used the validated SRQ with at least 6 score, as validated in Indonesian language (Isfandari, 2009). 

Table 1 To "clean up" the table and make it more intelligible, I suggest removing the N from the categories, informing only the total N of the variable and, for the categories, only the respective percentages.

Response:

Thank you. We deleted the N from the categories

Lines 180 to 181: "Physiological factors related to stunting were child sex (boy), low maternal MUAC, older maternal and paternal age, paternal and maternal short stature." As already mentioned, these variables are not physiological.

Response:

We stated those variables as physiological factors by merging the physiological factors for both child and parents. It is true that MUAC, BMI, and height was anthropometric measurement. Anthropometric measurements are a series of quantitative measurements of the muscle, bone, and adipose tissue used to assess the composition of the body (Casadei and Kiel, 2020). Thus, we categorized those variables as physiological factors.

Lines 260 to 262: "Based on the population of Indonesia in 2013, this would account for approximately 14.65 million stunted children. And 51.6% of under-five children experienced a parent with any level of distress- either maternal (39.6%) or paternal (33.3%) or both (21.3%);" These values are overestimated. In the first case, due to the inclusion of a high proportion of individuals distributed following the expected frequency in a healthy population and, in the case of distress, by the use of criteria in disagreement with what was proposed in the original method.

Response:

We deleted all any parental distress in this article, edited as follows (line 299-314). 

“The present study found that parental distress was associated with reduced linear growth of their children. More than half of under-five children were mildly, moderately, or severely stunted. Based on the population projection of Indonesia in 2013 there were approximately 23,994,200 under-five children (Statistics Indonesia, 2013), this would account for 15.31 million stunted children. And 7.5% of under-five children experienced parental distress - either maternal (4.1%) or paternal (2.6%) or both (0.9%). Of those children exposed to parental distress, approximately two-thirds (70.4%) had mild, moderate, or severe stunting. Based on these data we estimate approximately 1.27 million stunted children in Indonesia reside in households experiencing parental distress. This is a substantial burden that is likely to be underestimated. We note the proportion of households with psychological distress from the Self Reporting Questionnaire (SRQ20) at the provincial level ranged from 1.6% to 15.9%, with urban centers tending toward higher stress levels, suggesting this burden is likely to increase as urbanization is increasing globally. Under-five children are in a critical stage of development, needing stimulation and attention (Prado et al., 2017, McDonald et al., 2016). Distress among parents may affect parenting quality, especially responsiveness to their interactions, or exacerbate toxic stress that is known to affect child growth and development (DiPietro et al., 2006, Kiernan and Huerta, 2008).”

Lines 276 to 277: "Paternal and parental distress were associated with reductions of HAZ by 0.13, and 0.23 z-scores, respectively." Usually, mothers are the main responsible for the care of the child. In the analyzed sample, this should be made even more evident by the fact that the vast majority of fathers, unlike mothers, have work outside the home. So how explain that paternal distress contributes to growth deficit, but maternal distress does not?

Response:

We have edited the paragraph as follows (line 264-275).

" In this study, we found that for each one-unit increase in the score for paternal and parental distress, the HAZ of the child decreased by 0.11 and 0.19, respectively. This reduction is higher than the HAZ reduction related to maternal distress alone. Paternal distress has been commonly known to be related to low income and poverty, job insecurity, and social inequalities (World Health Organization, 2012). We were unable to find a previous study that yielded the relationship between parental and paternal distress on stunting or growth of children. Our study also indicates that maternal distress was associated with the reduction of HAZ by 0.09, and this is larger than previously reported by other studies in rural Bangladesh where infants at age 12 months whose mothers had symptoms of depression had a 0.007 drop in mean HAZ (Black et al., 2009). A longitudinal study of a nationally representative US birth cohort showed mothers with mild and moderate depressive symptoms had children who were, on average, 0.26 cm shorter in stature (95% CI: -0.48 cm to -0.05 cm) than their peers (Surkan et al., 2014)”

Lines 320 to 322: "We recommend promotion of mental and behavioral health for parents, which would promote child growth and also promote early childhood cognitive development". What would be the causes of parental distress? So, what actions should be taken to address this problem?

Response:

Thank you for bringing more detailed. The causes of parental distress could be at the individual attributes level, social circumstances, and environmental factors so that the actions taken (World Health Organization, 2012). We added in the paragraph some the actions taken as follows (line 335-340): 

“We recommend promotion of mental and behavioral health for parents, which would promote child growth and also promote early childhood cognitive development. The actions taken for mental health could be for developing and protecting individual attributes, supporting households and communities, and supporting vulnerable groups in society (World Health Organization, 2012). While the actions for behavioral health promotion for parent would be integrated in nutrition specific and sensitive activities in the stunting reduction system (United Nations Children's Fund, 2015).”

Lines 323 to 326: "Perhaps most striking is the observation that parental distress interacting with multiple other factors accounted for 5.6% of the overall loss of HAZ-score, virtually all of its effect, rendering it amongst the larger factors in poor linear growth in this analysis. This highlights the large contribution of psychosocial factors to linear growth restriction via other factors". This argument is difficult to understand: is not 5.6% the part of the loss in HAZ explained by the parental distress? If so, the last sentence does not exaggerate the importance given to distress, given that 94.4% of the growth restriction would be explained by other factors? Please clarify.

Response:

Thank you. We reanalyze using cut off point distress score >6. We edited the paragraph (341-345).

“We note that while parental distress directly accounted for only 0.6% of the overall loss of HAZ-score, its indirect effects are substantial given its role as risk factors for poor parenting (Shay et al., 2020, Frith et al., 2009), food security and consumption (Berge et al., 2020, Parks et al., 2012), morbidity (Ross et al., 2011), and health care access (Brown et al., 2011). Also, children must live in a safe, supportive, and nurturing family. Parental distress could interfere with the right of a child (Black et al., 2020). Other findings from this study are also important.”

Lines 336 to 340: "Behavior modification interventions should improve child growth by targeting improved waste water disposal, solid waste disposal, septic tank use, handwashing, and quitting smoking. While intervention on social conditions include increasing household income, improving maternal education, providing jobs, and attention to rural mothers' conditions and working mothers." Wouldn't these actions, in addition to preventing growth deficit, also be effective in reducing the prevalence of distress?

Response:

Yes. We agree that the improvement of social condition could also be effective in reducing the prevalence of distress. We edited the paragraph as follow (line 353-357):

“Behavior modification interventions should improve child growth by targeting improved water quality, garbage disposal, washing hand, and quitting smoking. While intervention on social conditions include increasing household income, improving maternal education, providing jobs, and attention to rural mothers’ conditions and working mothers. These interventions could also support reducing the prevalence of distress (World Health Organization, 2012).”

Lines 353 to 358: "Concerns regarding maternal distress in relation to child growth have been rising, and a better understanding is needed of interventions for maternal, paternal and parental distress. This would include more studies to understand how maternal, paternal and parental distress could reduce linear growth of the children, and therefore exacerbate the poor growth of the children. If this finding is confirmed, promotion of mental and behavioral health programs must to be pursued as part of a comprehensive strategy to enhance child growth and development." In addition to the harm in caring for the child, would any pathophysiological mechanisms be interfering with the growth process? In the introduction, the participation of the hypothalamic-pituitary-adrenal axis and other pathways, and changes in nutrient metabolism or immune function, was mentioned. Wouldn't it be worth discussing a little more about these mechanisms?

Response:

Thank you. We added in the discussion section as follows (line 373-383):

“Concerns regarding maternal distress in relation to child growth have been rising, and a better understanding is needed of interventions for maternal, paternal and parental distress. This would include more studies to understand how maternal, paternal and parental distress could reduce linear growth of the children, and therefore exacerbate the poor growth of the children. Psychological distress among parent could increase stress for the child. It could activate stress response through HPA axis to change intestinal permeability, motility, and mucus production. The HPA axis activation also affects immune cell responses (Fung et al., 2017). If this finding is confirmed, promotion of mental and behavioral health programs must to be pursued as part of a comprehensive strategy to enhance child growth and development, i.e. improve stakeholder capacity, integrated community development, improve parenting skills, as well as reduce gender discrimination, and domestic violence.”

Lines 364 to 368: "As such, while the cumulative loss of height contributed from parental distress itself was limited, the effects of parental distress on overall linear growth via indirect effects is substantial. Given that multiple behavioral and socioeconomic factors influence stunting, we suggest promotion of mental and behavioral health programs for parents to promote child growth and child development." Then, wouldn't it be more suitable to pay attention to the causes here called of indirect effects? Based on the reported findings, it would not be more appropriate to propose broader and more integrated actions on all determinants or, at least, relativize more prudently the role of distress? Caution is needed regarding the external validity of the study. The outcomes may vary according to the characteristics of the scenarios where the studies are developed. By the way, let's see the conclusion presented in the study by Harpham et al. [6]: "There was a relation between high maternal CMD and poor child nutritional status in India and Vietnam. However, the findings from Peru and Ethiopia do not provide clear evidence for a similar association being present in non-Asian countries. Regardless of the direction of the relation, child nutrition programmes in Asia should consider incorporating promotion of maternal mental health."

Response:

Thank you. We edited (line 391-395).

“Given that multiple behavioral and socioeconomic factors influence stunting, we suggest promotion of mental and behavioral health programs for parents must to be pursued as part of a comprehensive strategy to enhance child growth and child development, i.e, improve stakeholder capacity, integrated community development, improve parenting skills, as well as reduce gender discrimination, and domestic violence.”

1. Carvalhaes MA, Benicio MH: [Mother's ability of childcare and children malnutrition]. Revista de saude publica 2002, 36(2):188-197.

2. Girma S, Fikadu T, Abdisa E: Maternal Common Mental Disorder as Predictors of Stunting among Children Aged 6-59 Months in Western Ethiopia: A Case-Control Study. International journal of pediatrics 2019, 2019:4716482.

3. Hassan BK, Werneck GL, Hasselmann MH: Maternal mental health and nutritional status of six-month-old infants. Revista de saude publica 2016, 50:7.

4. Nguyen PH, Saha KK, Ali D, Menon P, Manohar S, Mai LT, Rawat R, Ruel MT: Maternal mental health is associated with child undernutrition and illness in Bangladesh, Vietnam and Ethiopia. Public health nutrition 2014, 17(6):1318-1327.

5. Santos DS, Santos DN, Silva Rde C, Hasselmann MH, Barreto ML: Maternal common mental disorders and malnutrition in children: a case-control study. Social psychiatry and psychiatric epidemiology 2011, 46(7):543-548.

6. Harpham T, Huttly S, De Silva MJ, Abramsky T: Maternal mental health and child nutritional status in four developing countries. Journal of epidemiology and community health 2005, 59(12):1060-1064.

7. Meza JM: [Prevalence of mental health problems in parents of children with malnutrition in one area of Tegucigalpa, Honduras]. Acta psiquiatrica y psicologica de America latina 1988, 34(2):145-148.

8. United Nations Children's Fund: Strategy for improved nutrition of children and women in developing countries. The Indian Journal of Pediatrics 1991, 58(1):13-24.

6. PLOS authors have the option to publish the peer review history of their article (what does this mean?). If published, this will include your full peer review and any attached files.

Do you want your identity to be public for this peer review? For information about this choice, including consent withdrawal, please see our Privacy Policy.

Reviewer #1: Yes: Isabel Giacomini Marques

Reviewer #2: Yes: HAROLDO DA SILVA FERREIRA

Response:

We have upload the figuresto the PACE and converted to the valid tiff

References

Agho, K. E., Akombi, B. J., Ferdous, A. J., Mbugua, I. & Kamara, J. K. 2019. Childhood undernutrition in three disadvantaged East African Districts: a multinomial analysis. BMC Pediatrics, 19, 118.

Beal, T., Tumilowicz, A., Sutrisna, A., Izwardy, D. & Neufeld, L. M. 2018. A review of child stunting determinants in Indonesia. Maternal & child nutrition, 14, e12617.

Benzeval, M. 1998. The self-reported health status of lone parents. Social science & medicine, 46, 1337-1353.

Berge, J. M., Fertig, A. R., Trofholz, A., Neumark-Sztainer, D., Rogers, E. & Loth, K. 2020. Associations between parental stress, parent feeding practices, and child eating behaviors within the context of food insecurity. Preventive Medicine Reports, 19, 101146.

Bham, G. H., Javvadi, B. S. & Manepalli, U. R. 2012. Multinomial logistic regression model for single-vehicle and multivehicle collisions on urban US highways in Arkansas. Journal of Transportation Engineering, 138, 786-797.

Bhutta, Z. A., Berkley, J. A., Bandsma, R. H. J., Kerac, M., Trehan, I. & Briend, A. 2017. Severe childhood malnutrition. Nature Reviews Disease Primers, 3, 17067.

Black, M. M., Baqui, A. H., Zaman, K., El Arifeen, S. & Black, R. E. 2009. Maternal depressive symptoms and infant growth in rural Bangladesh. The American Journal of Clinical Nutrition, 89, 951S-957S.

Black, M. M., Lutter, C. K. & Trude, A. C. 2020. All children surviving and thriving: re-envisioning UNICEF's conceptual framework of malnutrition. The Lancet Global Health, 8, e766-e767.

Black, R. E., Allen, L. H., Bhutta, Z. A., Caulfield, L. E., De Onis, M., Ezzati, M., Mathers, C. & Rivera, J. 2008. Maternal and child undernutrition: global and regional exposures and health consequences. The Lancet, 371, 243-260.

Brown, S. J., Yelland, J. S., Sutherland, G. A., Baghurst, P. A. & Robinson, J. S. 2011. Stressful life events, social health issues and low birthweight in an Australian population-based birth cohort: challenges and opportunities in antenatal care. BMC Public Health, 11, 196.

Camey, S. A., Torman, V. B. L., Hirakata, V. N., Cortes, R. X. & Vigo, A. 2014. Bias of using odds ratio estimates in multinomial logistic regressions to estimate relative risk or prevalence ratio and alternatives. Cadernos de saude publica, 30, 21-29.

Casadei, K. & Kiel, J. 2020. Anthropometric Measurement. StatPearls [Internet]. StatPearls Publishing.

Danaei, G., Andrews, K. G., Sudfeld, C. R., Fink, G., Mccoy, D. C., Peet, E., Sania, A., Fawzi, M. C. S., Ezzati, M. & Fawzi, W. W. 2016. Risk Factors for Childhood Stunting in 137 Developing Countries: A Comparative Risk Assessment Analysis at Global, Regional, and Country Levels. PLoS Med, 13, e1002164.

Das-Smaal, E. A. 1990. Biases in categorization. Advances in psychology. Elsevier.

Dipietro, J. A., Novak, M. F., Costigan, K. A., Atella, L. D. & Reusing, S. P. 2006. Maternal psychological distress during pregnancy in relation to child development at age two. Child development, 77, 573-587.

Field, T., Diego, M. & Hernandez-Reif, M. 2006. Prenatal depression effects on the fetus and newborn: a review. Infant Behavior and Development, 29, 445-455.

Frith, A. L., Naved, R. T., Ekström, E.-C., Rasmussen, K. M. & Frongillo, E. A. 2009. Micronutrient supplementation affects maternal-infant feeding interactions and maternal distress in Bangladesh. The American Journal of Clinical Nutrition, 90, 141-148.

Fund, U. N. C. S. 1991. Strategy for improved nutrition of children and women in developing countries. The Indian Journal of Pediatrics, 58, 13-24.

Fung, T. C., Olson, C. A. & Hsiao, E. Y. 2017. Interactions between the microbiota, immune and nervous systems in health and disease. Nat Neurosci, 20, 145-155.

Hair, J. F., Black, W. C., Babin, B. J. & Anderson, R. E. 2014. Multivariate data analysis: Pearson new international edition. Essex: Pearson Education Limited.

Isfandari, S. 2009. VALIDITAS DAN RELIABILITAS ITEM DISABILITAS DALAM RISET KESEHATAN DASAR 2007. Buletin Penelitian Kesehatan.

Jané-Llopis, E., Barry, M., Hosman, C. & Patel, V. 2005. Mental health promotion works: a review. Promotion & Education, 12, 9-25.

Kiernan, K. E. & Huerta, M. C. 2008. Economic deprivation, maternal depression, parenting and children's cognitive and emotional development in early childhood 1. The British journal of sociology, 59, 783-806.

Lv, J., Liu, Q., Ren, Y., Gong, T., Wang, S., Li, L. & Collaboration, C. I. F. H. 2011. Socio-demographic association of multiple modifiable lifestyle risk factors and their clustering in a representative urban population of adults: a cross-sectional study in Hangzhou, China. International Journal of Behavioral Nutrition and Physical Activity, 8, 40.

Mcdonald, S., Kehler, H., Bayrampour, H., Fraser-Lee, N. & Tough, S. 2016. Risk and protective factors in early child development: Results from the All Our Babies (AOB) pregnancy cohort. Research in developmental disabilities, 58, 20-30.

Nguyen, P. H., Friedman, J., Kak, M., Menon, P. & Alderman, H. 2018. Maternal depressive symptoms are negatively associated with child growth and development: Evidence from rural I ndia. Maternal & Child Nutrition, e12621.

Organization, W. H. 1986. Ottawa charter for health promotion. Health promotion, 1, iii-v.

Parks, E. P., Kumanyika, S., Moore, R. H., Stettler, N., Wrotniak, B. H. & Kazak, A. 2012. Influence of stress in parents on child obesity and related behaviors. Pediatrics, 130, e1096-104.

Pelletier, D. L., Frongillo Jr, E. A., Schroeder, D. G. & Habicht, J.-P. 1995. The effects of malnutrition on child mortality in developing countries. Bulletin of the World Health Organization, 73, 443.

Prado, E. L., Sebayang, S. K., Apriatni, M., Adawiyah, S. R., Hidayati, N., Islamiyah, A., Siddiq, S., Harefa, B., Lum, J., Alcock, K. J., Ullman, M. T., Muadz, H. & Shankar, A. H. 2017. Maternal multiple micronutrient supplementation and other biomedical and socioenvironmental influences on children's cognition at age 9-12 years in Indonesia: follow-up of the SUMMIT randomised trial. The Lancet Global Health, 5, e217-e228.

Rakotomanana, H., Gates, G. E., Hildebrand, D. & Stoecker, B. J. 2016. Determinants of stunting in children under 5 years in Madagascar. Maternal & Child Nutrition, n/a-n/a.

Ross, J., Hanlon, C., Medhin, G., Alem, A., Tesfaye, F., Worku, B., Dewey, M., Patel, V. & Prince, M. 2011. Perinatal mental distress and infant morbidity in Ethiopia: a cohort study. Arch Dis Child Fetal Neonatal Ed, 96, F59-64.

Shadfar, S. & Malekmohammadi, I. 2013. Structuring State Intervention Policies to Boost Rice Production by Multinomial Logistic and Ordinal Regression Application and Multicollinearity Cautiousness. Editorial Team, 123.

Shay, M., Tomfohr-Madsen, L. & Tough, S. 2020. Maternal psychological distress and child weight at 24 months: investigating indirect effects through breastfeeding in the All Our Families cohort. Canadian Journal of Public Health, 1-12.

Statistics Indonesia 2013. Indonesia population projection 2010-2035. Statistics Indonesia.

Stevens, G. A., Finucane, M. M., Paciorek, C. J., Flaxman, S. R., White, R. A., Donner, A. J. & Ezzati, M. 2012. Trends in mild, moderate, and severe stunting and underweight, and progress towards MDG 1 in 141 developing countries: a systematic analysis of population representative data. The Lancet, 380, 824-834.

Stewart, R. C. 2007. Maternal depression and infant growth–a review of recent evidence. Maternal & Child Nutrition, 3, 94-107.

Surkan, P. J., Ettinger, A. K., Hock, R., Ahmed, S., Strobino, D. M. & Minkovitz, C. S. 2014. Early maternal depressive symptoms and child growth trajectories: a longitudinal analysis of a nationally representative US birth cohort. BMC Pediatrics, 14, 185.

United Nations Children's Fund 2015. UNICEF's approach to scaling up nutrition for mothers and their children. Discussion Paper. Programme Division, UNICEF New York, NY.

Who Multicentre Growth Reference Study Group 2006. WHO Child Growth Standards: Length/height-for-age, weight-for-age, weight-for-age, weight-for-length, weight-for-height and body mass index-for-age: methods and development. Geneva: World Health Organization.

World Health Organization 2012. Risks to mental health: An overview of vulnerabilities and risk factors. Geneva: WHO.

World Health Organization 2019. Nutrition Landscape Information System (NLIS) country profile indicators: interpretation guide.

We have made every attempt to incorporate these suggestions as thoroughly as possible.

Thank you again for your interest in our work. We await your review of our revised manuscript.

Sincerely yours,

Kun A Susiloretni

---

## [Decision Letter · Decision Letter 1]

11 May 2021

PONE-D-20-08335R1

The psychological distress of parents is associated with reduced linear growth of children: evidence from a nationwide population survey

PLOS ONE

Dear Dr. Susiloretni,

Thank you for submitting your manuscript to PLOS ONE. After careful consideration, we feel that it has merit but does not fully meet PLOS ONE’s publication criteria as it currently stands. Therefore, we invite you to submit a revised version of the manuscript that addresses the points raised during the review process.

After initial aceptance, the authors have made additional changes in the revised manuscript. We ask the authors to submit the revised manuscript with essential changes or corrections with apropriate justification for each additional changes. Please keep the Figure 5 as presented in the previous submission. 

We look forward to receiving your revised manuscript.

Kind regards,

Marly A. Cardoso, Ph.D.

Academic Editor

PLOS ONE

Journal Requirements:

Reviewers' comments:

Reviewer's Responses to Questions

**Comments to the Author**

1. If the authors have adequately addressed your comments raised in a previous round of review and you feel that this manuscript is now acceptable for publication, you may indicate that here to bypass the “Comments to the Author” section, enter your conflict of interest statement in the “Confidential to Editor” section, and submit your "Accept" recommendation.

Reviewer #1: All comments have been addressed

2. Is the manuscript technically sound, and do the data support the conclusions?

Reviewer #1: Yes

3. Has the statistical analysis been performed appropriately and rigorously? 

Reviewer #1: Yes

4. Have the authors made all data underlying the findings in their manuscript fully available?

Reviewer #1: Yes

5. Is the manuscript presented in an intelligible fashion and written in standard English?

Reviewer #1: Yes

6. Review Comments to the Author

Reviewer #1: I would like to thank the authors for addressing all the comments. The modifications truly improved the manuscript and it should be accepted.

7. PLOS authors have the option to publish the peer review history of their article (what does this mean?). If published, this will include your full peer review and any attached files.

Reviewer #1: **Yes: **Isabel Giacomini

---

## [Author Response · Author response to Decision Letter 1]

4 Jul 2021

Dear Reviewer Dr. Isabel Giacomini,

Thank you very much for the supportive statement. We hope it will go to the production process.

Best regards,

Kun Susiloretni

6. Review Comments to the Author

Reviewer #1: I would like to thank the authors for addressing all the comments. The modifications truly improved the manuscript and it should be accepted.

Response:

Thank you very much for the supportive statement from the reviewer.

---

## [Editor Report · Decision Letter 2]

9 Jul 2021

PONE-D-20-08335R2

The psychological distress of parents is associated with reduced linear growth of children: evidence from a nationwide population survey

PLOS ONE

Dear Dr. Susiloretni,

Thank you for submitting your manuscript to PLOS ONE. After careful consideration, we feel that it has merit but does not fully meet PLOS ONE’s publication criteria as it currently stands. Therefore, we invite you to submit a revised version of the manuscript that addresses the points raised during the review process.

Please note the request to describe and justify the changes made after the acceptance of the manuscript, as previously pointed out: "After initial acceptance, the authors have made additional changes in the revised manuscript. We ask the authors to submit the revised manuscript with essential changes or corrections with apropriate justification for each additional changes. Please keep the Figure 5 as presented in the previous submission." Please state this in the replay letter.  

We look forward to receiving your revised manuscript.

Kind regards,

Marly A. Cardoso, Ph.D.

Academic Editor

PLOS ONE

Journal Requirements:

Additional Editor Comments (if provided):

Please note the request to describe and justify the changes made after the acceptance of the manuscript, as previously pointed out: "After initial acceptance, the authors have made additional changes in the revised manuscript. We ask the authors to submit the revised manuscript with essential changes or corrections with apropriate justification for each additional changes. Please keep the Figure 5 as presented in the previous submission." Please state this in the replay letter.

---

## [Author Response · Author response to Decision Letter 2]

22 Aug 2021

Dear Prof. Marly A. Cardoso, Ph.D.

Academic Editor of PLOS ONE

RE: PONE-D-20-08335R2

The psychological distress of parents is associated with reduced linear growth of children: evidence from a nationwide population survey 

We appreciate the time and effort dedicated to providing the feedback. Thank you for giving us the opportunity to revise our manuscript in hopes it will merit publication in PLOS ONE. 

After initial acceptance, we have made additional changes in the revised manuscript. The editor asked the authors to submit the revised manuscript with essential changes or corrections with appropriate justification for each additional change. 

We gave given some explanations to describe and justify the changes made after initial acceptance of the manuscript. We also reverted to the original Figure 5 as presented in the previous submission.

We hope the following responses and the corresponding revision of the manuscript fulfills the Editor’s and Reviewer’s requirements for considering this manuscript for publication. 

Thank you again for your consideration, and I look forward to hearing from you.

Best regards,

Kun A Susiloretni

Dear Dr. Susiloretni,

Thank you for submitting your manuscript to PLOS ONE. After careful consideration, we feel that it has merit but does not fully meet PLOS ONE’s publication criteria as it currently stands. Therefore, we invite you to submit a revised version of the manuscript that addresses the points raised during the review process.

Please note the request to describe and justify the changes made after the acceptance of the manuscript, as previously pointed out: "After initial acceptance, the authors have made additional changes in the revised manuscript. We ask the authors to submit the revised manuscript with essential changes or corrections with apropriate justification for each additional changes. Please keep the Figure 5 as presented in the previous submission." Please state this in the replay letter. 

We look forward to receiving your revised manuscript.

Kind regards,

Marly A. Cardoso, Ph.D.

Academic Editor

PLOS ONE

Journal Requirements:

Response:

We have checked the reference list to ensure that it is complete and correct. And there were no papers that have been retracted.

Additional Editor Comments (if provided):

Please note the request to describe and justify the changes made after the acceptance of the manuscript, as previously pointed out: "After initial acceptance, the authors have made additional changes in the revised manuscript. We ask the authors to submit the revised manuscript with essential changes or corrections with apropriate justification for each additional changes. Please keep the Figure 5 as presented in the previous submission." Please state this in the replay letter.

Response:

We gave some explanations to describe and justify the changes made after initial acceptance of the manuscript as below. We have also reverted to the original Figure 5 as requested by the editors. 

Here with are the explanation of the changes.

Lines 37-54. We have made some edits to improve clarity and flow 

Line 55: We added the mention of the risk factor of paternal occupational status as it was significant in the analysis but we had missed its inclusion in the previous abstract.

Lines 60-61: We added this sentence to show the main finding of the paper.

Lines 70-129: We have edited the text to make it more clear, correct, and unambiguous. 

Lines 129-130: We added this to confirm that all children living in a household were included in the analysis.

Lines-138-151: We have rephrased this part to improve flow and comprehension. 

Lines 156-176: Again, we edited several sentences, in order to improve clarity and flow.

Lines 182-184: We have added the explanation of reason why we used PSU as a random effect, and not household. 

Lines 186-191. We have rephrased to make it more intelligible and with better flow. 

Lines 227-309. We have edited the text to provide more clear meaning that is correct, and unambiguous. 

Lines 284-287. We created consistency so that all estimations are followed by confidence intervals and p values.

Lines 310-318. We rearranged the text and explained the association of distress with multiple risk factors. This was done to explain in more detail that distress has association with other risk factors. 

Lines 319-330. We added the percentage of HAZ-score lost of each factor to enable the readers to simultaneously grasp these percentages. 

Line 331-338. We have added the percentage of children reside in a household with any risk factor, and added the percentage of children who could be affected by the risk factors of parental distress. We did this to show the proportion of populations wherein enhanced parental stress may be problematic.

Lines 343-474. In the discussion section, we have edited and rearranged the text in order to improve clarity and flow, and also to present it in a more intelligible fashion.

Lines 477-487. In the conclusions, we have edited the text in order to improve clarity and flow.

All tables. We made them more consistent by putting units in bracket following each risk factor.

All figure titles. We edited all figure captions to make the more clear in meaning.

---

## [Decision Letter · Decision Letter 3]

9 Sep 2021

The psychological distress of parents is associated with reduced linear growth of children: evidence from a nationwide population survey

PONE-D-20-08335R3

Dear Dr. Susiloretni,

We’re pleased to inform you that your manuscript has been judged scientifically suitable for publication and will be formally accepted for publication once it meets all outstanding technical requirements.

Kind regards,

Marly A. Cardoso, Ph.D.

Academic Editor

PLOS ONE

Additional Editor Comments (optional):

Reviewers' comments:

Reviewer's Responses to Questions

**Comments to the Author**

1. If the authors have adequately addressed your comments raised in a previous round of review and you feel that this manuscript is now acceptable for publication, you may indicate that here to bypass the “Comments to the Author” section, enter your conflict of interest statement in the “Confidential to Editor” section, and submit your "Accept" recommendation.

Reviewer #1: All comments have been addressed

2. Is the manuscript technically sound, and do the data support the conclusions?

Reviewer #1: Yes

3. Has the statistical analysis been performed appropriately and rigorously? 

Reviewer #1: Yes

4. Have the authors made all data underlying the findings in their manuscript fully available?

Reviewer #1: Yes

5. Is the manuscript presented in an intelligible fashion and written in standard English?

Reviewer #1: Yes

6. Review Comments to the Author

Reviewer #1: All my previous comments have been addressed by the authors. After the modifications, the text is now better organized, with fully available information about the methodology and findings. Overall, the manuscript is well written, leading to no misinterpretation or doubts.

7. PLOS authors have the option to publish the peer review history of their article (what does this mean?). If published, this will include your full peer review and any attached files.

Reviewer #1: **Yes: **Isabel Giacomini Marques

---

## [Editor Report · Acceptance letter]

16 Sep 2021

PONE-D-20-08335R3 

The psychological distress of parents is associated with reduced linear growth of children: evidence from a nationwide population survey 

Dear Dr. Susiloretni:

I'm pleased to inform you that your manuscript has been deemed suitable for publication in PLOS ONE. Congratulations! Your manuscript is now with our production department. 

Kind regards, 

on behalf of

Dr. Marly A. Cardoso 

Academic Editor

PLOS ONE